# Optogenetic control of a GEF of RhoA uncovers a signaling switch from retraction to protrusion

Jean de Seze, Maud Bongaerts, Benoit Boulevard, Mathieu Coppey*

Laboratoire Physico Chimie Curie, Institut Curie, PSL Research University, Sorbonne Université, Paris, France

## eLife Assessment

This **important** study combines **compelling** experiments with optogenetic actuation and **convincing** theory to understand how signalling proteins control the switch between cell protrusion and retraction, two essential processes in single cell migration. The authors examine the importance of the basal concentration and recruitment dynamics of a guanine exchange factor (GEF) on the activity of the downstream effectors RhoA and Cdc42, which control retraction and protrusion. The experimental and theoretical evidence provides a model of RhoA's involvement in both protrusion and retraction and shows that these complex processes are highly dependent on the concentration and activity dynamics of the components.

*For correspondence:
mathieu.coppey@curie.fr

Competing interest: The authors declare that no competing interests exist.

**Abstract** The ability of a single protein to trigger different functions is an assumed key feature of cell signaling, yet there are very few examples demonstrating it. Here, using an optogenetic tool to control membrane localization of RhoA nucleotide exchange factors (GEFs), we present a case where the same protein can trigger both protrusion and retraction when recruited to the plasma membrane, polarizing the cell in two opposite directions. We show that the basal concentration of the GEF prior to activation predicts the resulting phenotype. A low concentration leads to retraction, whereas a high concentration triggers protrusion. This unexpected protruding behavior arises from the simultaneous activation of Cdc42 by the GEF and sequestration of active RhoA by the GEF PH domain at high concentrations. We propose a minimal model that recapitulates the phenotypic switch, and we use its predictions to control the two phenotypes within selected cells by adjusting the frequency of light pulses. Our work exemplifies a unique case of control of antagonist phenotypes by a single protein that switches its function based on its concentration or dynamics of activity. It raises numerous open questions about the link between signaling protein and function, particularly in contexts where proteins are highly overexpressed, as often observed in cancer.

## Introduction

Cell protrusion and retraction are two morphological changes at the core of various cellular functions, including cell motility, adhesion, and tissue development. For example, during migration the cell must polarize itself by simultaneously controlling two opposite shape changes, protrusion at the front and retraction at the back (**Ridley et al., 2003**). Usually, these two morphodynamical events are thought to be spatially and temporally controlled by the segregation of protein activities or component within the cell space. Key controllers of these two processes are the members of the Rho family of small GTPases, in particular the three best-known RhoA, Rac1, and Cdc42 (**Jaffe and Hall, 2005**). These small proteins act as molecular switches, being turned on in their active GTP form by a large variety

of guanine exchange factors (GEFs) and turned off in the GDP form by GTPase activating proteins (GAPs) (*Cherfils and Zeghouf, 2013*). Early experiments expressing these GTPases in their active forms (*Hall, 1998*) led to the canonical picture in which Rac1 and Cdc42 promotes protrusion, with the presence of ruffles and filopodia, while RhoA promotes retraction, as testified by cell rounding and actomyosin contractility.

Yet, a large body of works on the regulation of GTPases has revealed a much more complex picture with numerous crosstalks and feedbacks allowing the fine spatiotemporal patterning of GTPase activities (*Pertz, 2010*) hereby questioning the legitimacy of the simple canonical picture. This is especially true for RhoA, since it has been proposed to be responsible for both protrusion and retraction depending on the cellular context (*Machacek et al., 2009*; *Pertz et al., 2006*; *Worthylake et al., 2001*). It thus remains to be addressed whether the induction of protrusions and retractions are simply set by the independent and segregated activities of GTPases or set by a more complicated signal integration.

In recent years, optogenetics has emerged as a very powerful tool to go beyond correlations and to demonstrate causality in this question. Up to now, the optogenetic approaches that have been developed to control GTPases in space and time tended to confirm the canonical picture. For example, we and others have shown that for Rac1 and Cdc42, despite of a complex cross-activation, the recruitment at the plasma membrane of minimal GEF activating domains causally induce cell protrusions (*de Beco et al., 2018*) and migration led by the front (*Vaidžiulytė et al., 2022*). Along the same line, optogenetic approaches to control RhoA, reviewed in *Chandrasekar et al., 2023* and *de Seze et al., 2023*, have all shown a causal induction of cell retraction.

Here, we report a serendipitous discovery where the optogenetic recruitment of guanine nucleotide exchange factors (GEFs) for RhoA to the plasma membrane triggers both protrusion and retraction in the same cell type, effectively polarizing the cell in opposite directions. Among the GEFs tested, PDZ-RhoGEF (PRG, also known as ARHGEF11) proved most effective at eliciting both phenotypes. We show that the outcome of this optogenetic perturbation depends on the basal concentration of the GEF prior to activation. At low concentrations, PRG recruitment induces cell retraction, consistent with the expected function of a RhoA GEF. However, at high concentrations, PRG activates Cdc42 and sequestrates active RhoA preventing its binding to effectors, leading to cell protrusion. Using a minimal mathematical model, we predict and confirm experimentally that modulating the frequency of PRG recruitment at intermediate concentrations can induce either phenotype. This ability to control opposing phenotypes on timescales of seconds exemplifies the multiplexing capacity of signaling circuits driven by a single protein.

## Results

### Local optogenetic recruitment to the plasma membrane of a DH–PH domain of RhoA GEF can lead to both protrusion and retraction in single cells

To control the activity of RhoA in migrating cells, we developed optogenetic tools based on the iLID-SspB light-gated dimerization system to recruit activating domains of GEFs specific to RhoA (*Figure 1A*). Using a strategy already applied to other GTPases (*Guntas et al., 2015*) and to RhoA itself (*Inaba et al., 2021*), we anchored the iLID part of the dimer to the membrane thanks to a CAAX motif together with the fluorescent protein mVenus, and expressed in the cytosol the DH–PH domains of different GEFs of RhoA fused to the SspB protein (*Figure 1B*). We cloned the iLID-SspB dimers into a plasmid separated by a P2A motif so that they were expressed at a one-to-one ratio. We selected three of the best-known GEFs of RhoA: LARG (ARHGEF12), GEF-H1 (ARHGEF2), and PDZ-RhoGEF (ARHGEF11). The LARG DH domain was already used with the iLid system (*Wagner and Glotzer, 2016*), the GEF-H1 DH domain had been recruited by optochemical methods (*Kamps et al., 2020*), and PRG was used with the CRY2-CIBN system (*Valon et al., 2017*). We chose to add the PH domains of GEFs in our constructs since it participates to an auto-amplification process (*Chen et al., 2010*) and appears to be sometimes required for GEF specificity (*Ju et al., 2022*). All strategies developed so far have been shown to trigger cell contractility, either in single cells or in monolayers.

We then transiently transfected RPE1 cells with our optogenetic systems (henceforth called optoPRG, optoLARG, and optoGEF-H1) and examined the effects of pulsatile local activation done

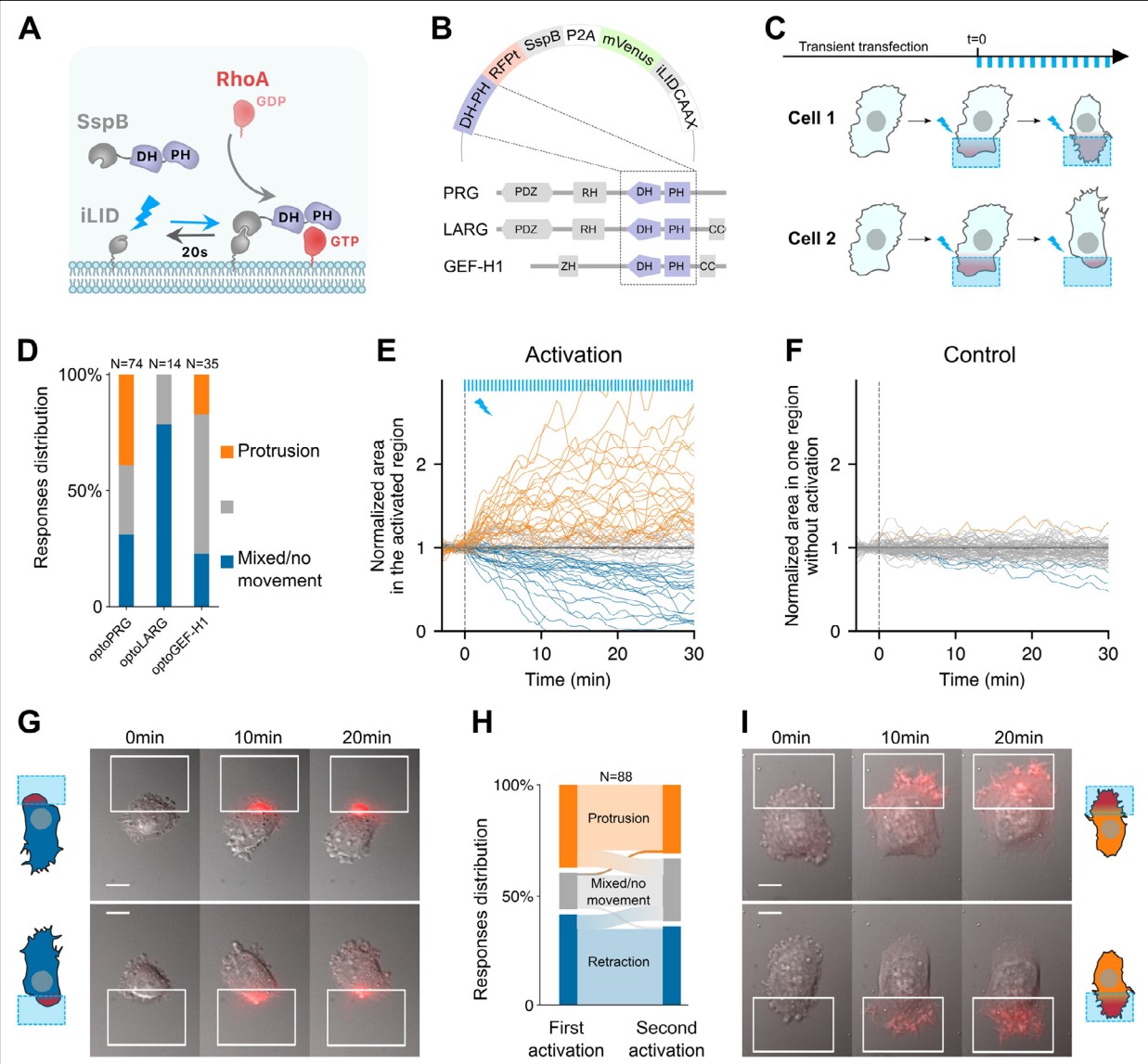

**Figure 1.** Optogenetic activation of RhoA leads to protrusion or retraction. (**A**) Scheme of the optogenetic tool. Optogenetic dimer (in gray) dimerizes upon blue light activation (blue arrow), and dissociates in the dark with an off rate of 20 s (black arrow). DH–PH domain is fused to the SspB moiety (purple), which recruitment to the plasma membrane through iLID triggers RhoA activation, from GDP (light red) to GTP (dark red) state. (**B**) The three opto plasmids, with DH–PH domains (purple) shown in their wildtype position in the different RhoA GEFs used here. (**C**) Experimental timeline. Transient transfection is done at least 30 hr before local activation (blue squares). Activation is done by pulses (blue bars, top) at different frequencies, intensities and durations. Cells are observed for 30–60 min. (**D**) Responses distributions for each optogenetic tool. Area over time of the cell in the activated region (**E**) or without activation (**F**), normalized by the mean initial area. $t = 0$ is the starting point of the activation, each blue bar on top representing one light impulse. Orange: protruding cells, blue: retracting cells, gray: nonmoving cells or mixed phenotype (labeled by hand). (**G, I**) Representative cells doing retraction (on the left) and protrusion (on the right) upon optogenetic activations on two different side of the cell. Scale bar: 10 µm. White squares: area of activation. Red color: RFPt channel (optogenetic tool). (**H**) Sankey diagram representing the proportion of cells doing a protrusion (orange), retraction (blue), or a mixed phenotype (gray) at one side (first activation) or the other side (second activation).

The online version of this article includes the following figure supplement(s) for figure 1:

**Figure supplement 1.** Mixed phenotype, cell polarity, and dual phenotype in another cell line.

thanks to a digital micromirror light source (see Materials and methods). In all cases, we saw a clear recruitment of the cytosolic part to the plasma membrane, between three- and tenfold, depending on the transfection intensity and the amount of light sent to the sample.

To our surprise, recruitment of DH–PH domains of the three GEFs of RhoA to the membrane with the same experimental procedure resulted in very different phenotypes (***Figure 1C, D*** and

*Videos 1–2*). Whereas optoLARG elicited only the expected retractile phenotype typical of RhoA pathway (*Video 2*), optoGEF-H1 and optoPRG exhibited less predictable behavior (*Figure 1D*). The retractile phenotype was observed in many cells, but a large proportion of cells showed a seemingly opposite response, namely a clear protruding phenotype with filopodia and ruffles (*Figure 1C*). For optoGEF-H1, most of the cells were very round after transfection and showed no response at all after optogenetic recruitment of the DH–PH domain to the membrane. Only few cells (~40%) showed distinct morphological changes after activation, most of them retracted, but some also showed a clear protruding phenotype (*Video 2*). For optoPRG, most cells showed a distinct phenotypic response upon blue light exposure (*Figure 1D*). Almost 35% of the cells exhibited a clear retracting phenotype leading to the formation of blebs or protrusions at the other side of the cell (*Video 1*), while 40% exhibited a markedly protruding phenotype (*Video 1*), reminiscent of the effect of an optoGEF for Cdc42 or Rac1 (*de Beco et al., 2018*) and often leading to retraction of the other non-activated pole of the cell. A small fraction of cells (~10%) showed no clear response, and the remaining cells (~25%) showed a mixed phenotype: while we could see ruffles or filopodia forming at the site of activation, the cell did not move and appeared to contract at the same time, leading to blebs at the other pole of the cell (*Figure 1—figure supplement 1A*).

To classify the different phenotypes in an unbiased manner, we computed the evolution of the membrane area inside the activation square during the activation with (*Figure 1E*) or without light (*Figure 1F*). We could see the clear impact of the recruitment of the optoPRG, that either triggers a diminution of membrane area – a retraction – or an increase in membrane area – a protrusion, while control cells show a much smaller membrane movement over the time course of the experiment. Quantification of the changes in membrane area in both the activated and non-activated part of the cell (*Figure 1—figure supplement 1B, C*) reveals that the whole cell is moving, polarizing in one direction or the other upon optogenetic activation.

To ensure that the cell response was not due to an already set polarity, we repeated the activation at the other pole of the cell 1 hr after the first round of activation, when cells had recovered from the first activation and were back to a resting state. Almost all cells retained the same phenotype: they protruded when they were already protruding and retracted when they were already retracting (*Figure 1G–I*). Only few cells showed a mixed phenotype in the first or second round, whereas their phenotype was clear for the other round (*Figure 1H*). Thus, we concluded that for one type of optogenetic activation (one frequency and one light intensity), the phenotype triggered by the recruitment of the optoPRG to the membrane was determined by the state of the cell and not by the cell section in which the optogenetic activation was performed.

To further verify that these opposite phenotypes were not cell line specific, we performed the experiment in Hela cells. The two phenotypes could be observed after transient transfection with optoPRG and activation with the same protocol (*Figure 1—figure supplement 1D, E*), showing that these two opposites phenotypes are not restricted to RPE1 cells only. Thus, the DH–PH domain of

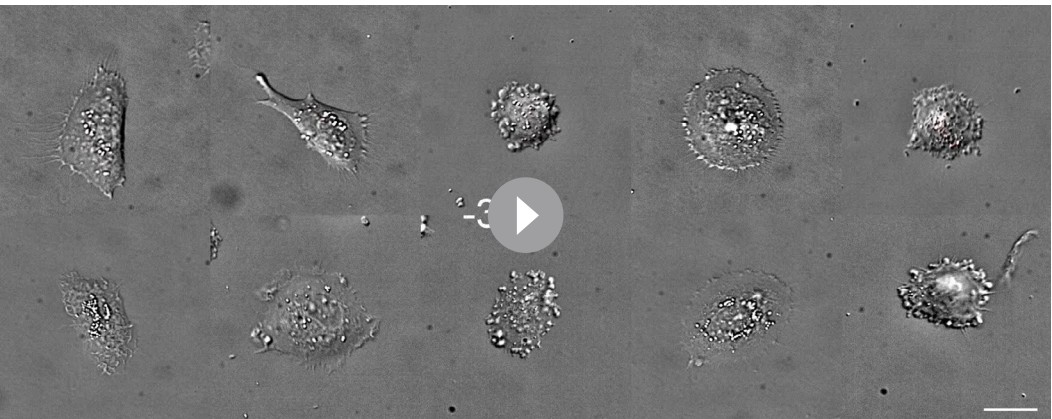

**Video 1.** Example of five protruding (top) and five retracting (bottom) cells activated with optoLARG (white box). Differential interference contrast (DIC) transmitted light overlayed with TIRFM optoLARG signal in red. Scale bar: 10 µm.

https://elifesciences.org/articles/93180/figures#video1

PRG can induce both retraction and protrusion upon its recruitment in single cells. These opposite phenotypes occur in the same cell line, in the same biomechanical environment, and with the same dynamics of recruitment to the membrane. This dual behavior is not cell specific or exclusive to this GEF of RhoA, as it has also been observed with GEF-H1, although with much less efficiency. For the rest of our study, we focused on optoPRG that elicits the strongest and most reproducible opposing phenotypes.

### Cell phenotype upon optogenetic activation depends on the cytosolic concentration of exogenous optoPRG

We next wonder what could differ in the activated cells that lead to the two opposite phenotypes. As all the experiments above were done by transiently transfecting cells, the cell-to-cell variability of expression led to a wide range of cytosolic concentrations of optoPRG that we can estimate by measuring the mean fluorescence intensity. Looking at the area in the activated region after 5 min against the cytosolic concentration of optoPRG leads to a very clear result (*Figure 2A*). Below a specific concentration threshold (~40 a.u.), almost all the activated cells (~95%) retract their membrane within the activation area, while above this threshold, ~85% of the cells extend their membrane (see *Figure 2B*

for selected examples). This result clearly demonstrates that the phenotype triggered by the activation of optoPRG can be predicted by its concentration within the cell before activation.

This leads to two hypotheses. Either the cell is responding to differences in the absolute amount of optoPRG recruited at the membrane, or the cell is in a different state before optogenetic activation that leads to opposite responses to optoPRG activation. To exclude one of these hypotheses, we first looked at the absolute recruitment at the membrane (*Figure 2C*). Even if the absolute recruitment depends on the initial concentration, we saw a lot of retracting cells reaching very high absolute optoPRG recruitment levels, which tends to exclude the first hypothesis. To further demonstrate it, we overexpressed in a different fluorescent channel a non-recruitable DH–PH domain of PRG, together with the optoPRG, in order to decouple global and recruited PRG concentration. We saw that overexpressing PRG DH–PH strongly increased the number of protruding phenotypes in low expressing cells (from 0% to 45%, *Figure 2A*, compare also with *Figure 2A*). Moreover, it was clear that the phenotype switch also correlates with the initial concentration of overexpressed PRG DH–PH domain, highly expressing cells being more prone to protrude (orange dots on the left, *Figure 2A*).

Most optogenetic tools that control GTPases activity rely on the fact that GEF domains are less active in the cytosol. Here, the phenotype switch is dictated by the amount of overexpressed optoPRG. Thus, PRG DH–PH domain must have some activity before optogenetic recruitment to the membrane, changing protein basal activities and/or concentrations – which we called the cell state. To confirm that cells were in a different state, we first measured the average

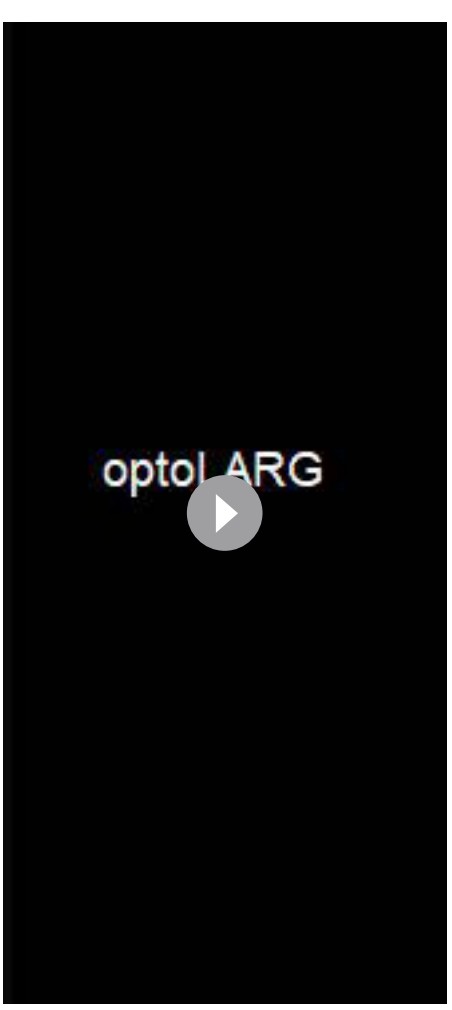

**Video 2.** Example of a blebbing cell upon optoLARG activation, and two examples of protruding and retracting cells upon optoGEFH1 activation (white box). Differential interference contrast (DIC) transmitted light overlayed with TIRFM optoLARG signal in red. Scale bar: 10 μm.

https://elifesciences.org/articles/93180/figures#video2

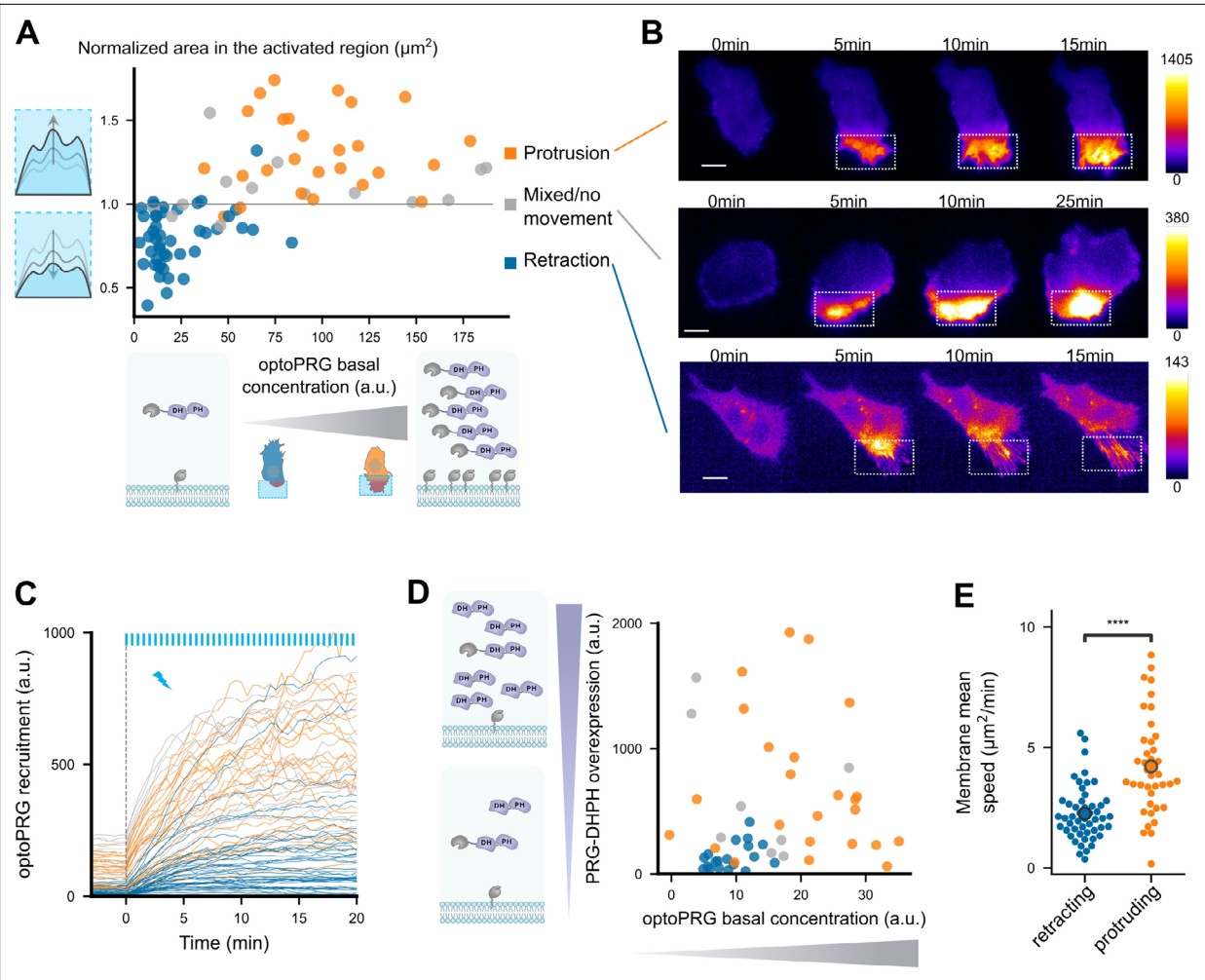

**Figure 2.** Cell phenotype depends on the initial optoPRG concentration. (**A**) Phenotype dependence on initial cytosolic optoPRG concentration. The normalized area in the activated region after 5 min is plotted against the mean fluorescence intensity for each cell, which color is labeled by hand depending on the observed phenotype. Schemes on the bottom represent high and low levels of expression (P2A plasmids implies approximately a one-to-one ratio of iLID against SspB). (**B**) Three representative time lapse images of transiently transfected cells, one retracting (top), one showing a mixed phenotype (middle), and one protruding (bottom). Intensities are very different, as seen by the dynamic range of the colormaps presented on the right. (**C**) Absolute fluorescence intensity of recruited optoPRG, before ($t < 0$) and after ($t > 0$) activation. Blue bars: activation pulses. (**D**) Phenotype depending on both optoPRG concentration and PRG DH–PH overexpression, measured both by fluorescence intensity (a.u.). Increasing recruitable and non-recruitable DH–PH domain of PRG both lead to protruding phenotypes. Phenotypes are manually labeled. (**E**) Membrane mean absolute displacement to compare membrane activity between retracting (blue) and protruding (orange) cells (Mann–Whitney *U* test. ****<0.0001).

The online version of this article includes the following figure supplement(s) for figure 2:

**Figure supplement 1.** OptoPRG expression leads to an increase of cell area and RhoA activity.

cell area before activation in the two phenotypes, and saw a strong and significant difference, highly expressing cells being 1.5 times bigger than low expressing cells (*Figure 2—figure supplement 1*). We then looked at membrane ruffles without any optogenetic activation, by calculating the mean absolute speed of membrane displacement in one region. We also observed a significant difference, protruding cells having more and stronger membrane displacements before being activated (*Figure 2E*).

This series of experiments led us to the conclusion that the main determining factor of the phenotype is the cytosolic concentration of the DH–PH domain of PRG before optogenetic activation. This initial concentration changes the cell state, in a way that the recruitment of the DH–PH domain of PRG triggers opposite cellular responses.

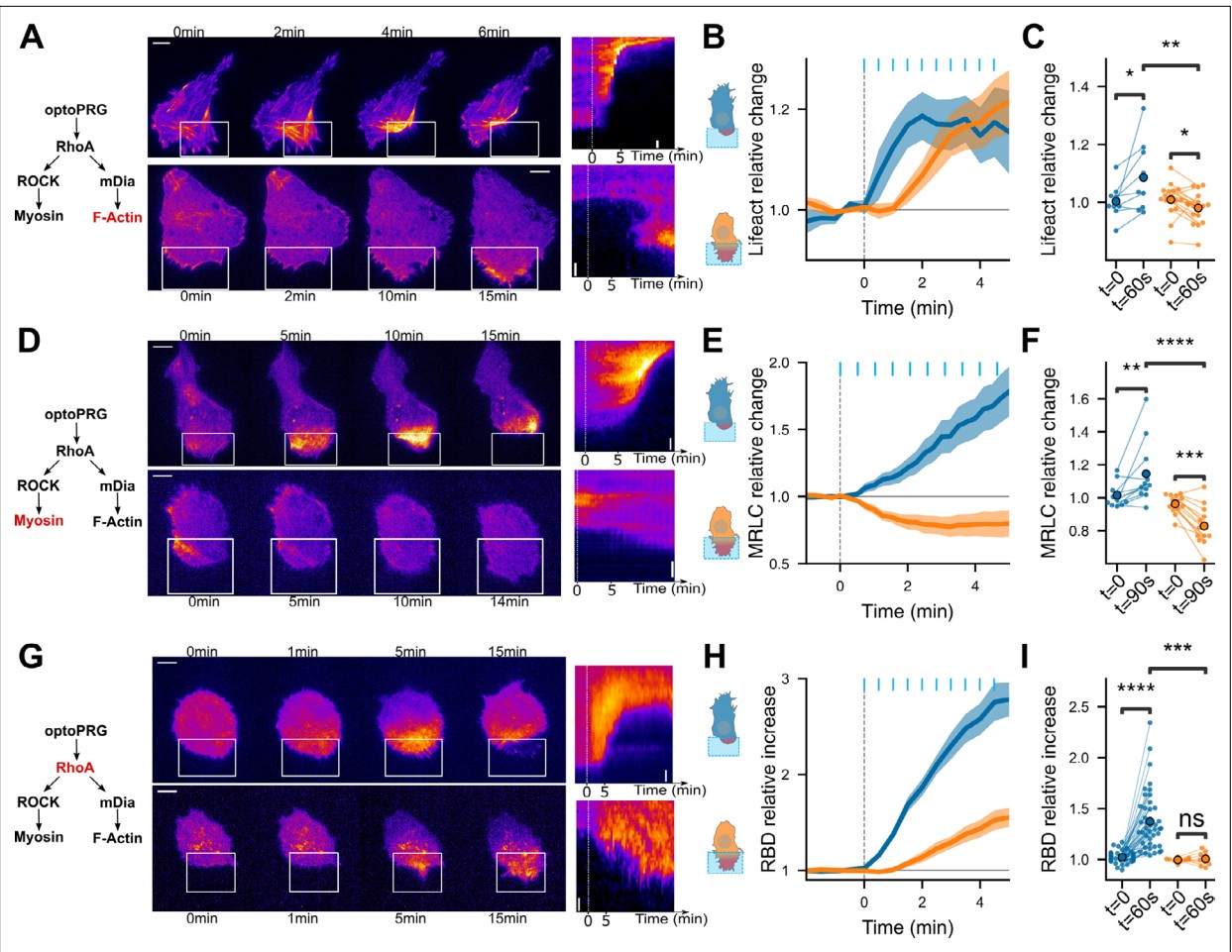

**Figure 3.** Downstream effectors show that cell phenotype is set immediately. Distinct pathways are triggered from the first timepoint. (**A,D,G**) Representative timelapse images and kymographs of retracting (top) and protruding (bottom) cells labeled with Lifeact-iRFP (**A**), MRLC-iRFP (**D**), and RBD-2xTdTomato biosensor (**G**), activated with optoPRG starting at $t = 0$ min. White rectangles are areas of optogenetic activation. Scale bars are 10 μm. (**B, E, H**) Corresponding mean normalized intensities are plotted against time (mean ± s.e.m.), blue for retracting cells and orange for protruding one. (**C, F, I**) Corresponding pairwise comparison for each cell of the signal inside the region of activation between the initial time and 60 s (Lifeact-iRFP and RhoA biosensor) or 90 s (MRCL-iRFP). Data are grouped by phenotype. *$p < 0.05$, **$p < 0.01$, ***$p < 0.001$, ****$p < 0.0001$ (Wilcoxon test to compare $t = 0$ and $t > 0$, independent $t$-test otherwise).

## Two distinct signaling pathways are triggered from the first activation timepoint

Very surprised by this ability of one protein to trigger opposite phenotypes, we sought to further characterize these responses by monitoring the activity of key proteins involved in the RhoA pathway. To this end, we initially investigated whether the variation in actin and myosin levels following optogenetic activation corresponded to the stereotypical dynamics of protrusions and retractions described in a previous study by *Martin et al., 2016*. We thus monitored Lifeact-iRFP and MRLC-iRFP (myosin regulatory light chain) proteins during the optogenetic experiments. In the retraction phenotype, the first recruitment of optoPRG to the membrane was followed by an immediate polymerization of actin in the area of activation, as we quantified over multiple experiments (*Figure 3A–C* and *Video 3*). Myosin recruitment appeared to follow the polymerization of actin, leading to the retraction of the membrane in the minutes following the activation (*Figure 3D–F*). In contrast, when cells were protruding, we observed a decrease in MRLC and Lifeact intensities following the first activation of our tool by light (*Figure 3A–F*), as it was previously shown for protrusions triggered by PDGF (*Martin et al., 2016*). This first decrease is probably due to the depolymerization of stress fibers at the area of activation. It is followed by an increase in actin polymerization, most probably responsible for the

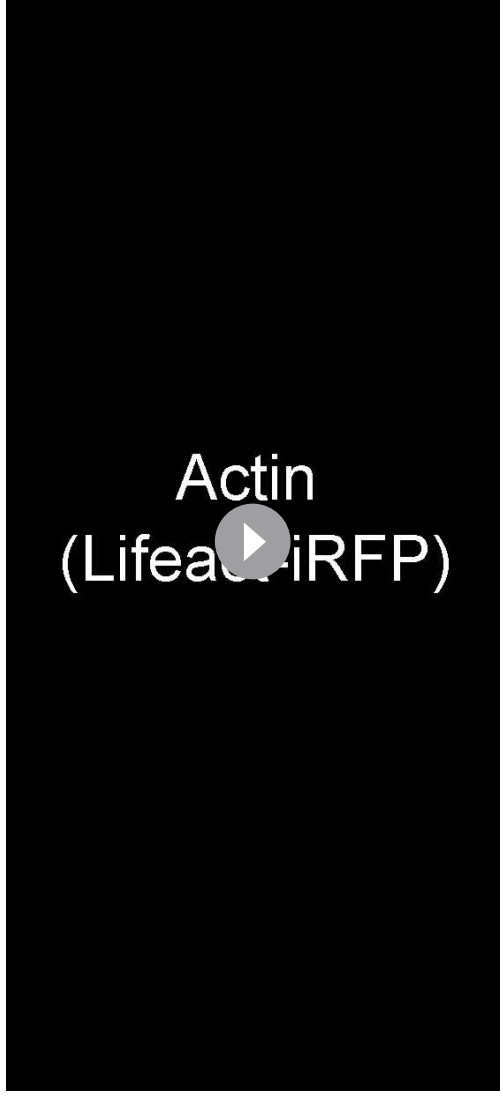

**Video 3.** Actin, myosin, and RhoA biosensor dynamics upon optoLARG activation (white box). Scale bar: 10 μm.
https://elifesciences.org/articles/93180/figures#video3

net displacement of the membrane, while myosin stays lower than the initial state. These differences in actin and myosin dynamics for the two phenotypes are visible at 30 s, one frame after the first pulse of light (*Figure 3C, F*), even if the cell phenotype cannot be distinguished by eye before 1 or 2 min. This shows that the pathways triggered by optoPRG recruitment differ from the first tens of seconds after light activation. We thus turned to RhoA itself, which is supposed to be upstream of actin and myosin, and just downstream of PRG. As our optogenetic tool prevented us from using FRET biosensors because of spectral overlap, we turned to a relocation biosensor that binds RhoA in its GTP form (*Mahlandt et al., 2021*). This highly sensitive biosensor is based on the multimeric TdTomato, whose spectrum overlaps with the RFPt fluorescent protein used for quantifying optoPRG recruitment. We thus designed a new optoPRG with iRFP, which could trigger both phenotypes but that turned out to be harder to transiently express at high levels, giving rise to a majority of retracting phenotype. Looking at the RhoA biosensor, we saw very different responses for both phenotypes (*Figure 3G–I* and *Video 3*). While retracting cells led to strong and immediate activation of RhoA, protruding cells showed a much smaller and delayed response. After the first minute of optogenetics activation, no significant change was seen in protruding cells (*Figure 3I*). Thus, either an unknown factor is sequestering RhoA-GDP, or optoPRG preferentially binds to another partner, both hypotheses being non-mutually exclusive.

Altogether, these data demonstrate that cells exhibit very different responses to optoPRG activation, regarding both actin polymerization and myosin activity, and even RhoA activity itself. They show that different pathways are immediately engaged, already at the level of Rho-GTPases, few tens of seconds after the first optogenetic activation.

## PH domain of PRG can inhibit RhoA signaling and is necessary for the protruding phenotype

As the phenotype triggered by optoPRG, revealed on a minute timescale, seems to be set by the reactions of the intracellular biochemical network after few tens of seconds, we turned to an analysis of RhoA activity at shorter timescale. Our first surprise came while looking at the response of the RhoA biosensor to pulses of optoPRG activation. We show in *Figure 4A*, three representative examples of such responses. While at low optoPRG concentration (cell 1), RhoA biosensor intensity follows the pulses of optoPRG with some delay; at higher concentrations (cells 2 and 3), RhoA biosensor intensity shows a very different behavior: it first decays, and then rises. It seems that, adding to the well-known activation of RhoA, PRG DH–PH can also functionally inhibit RhoA signaling.

Knowing that the PH domain of PRG triggers a positive feedback loop thanks to its binding to active RhoA (*Chen et al., 2010*), we hypothesized that this binding could sequester active RhoA at

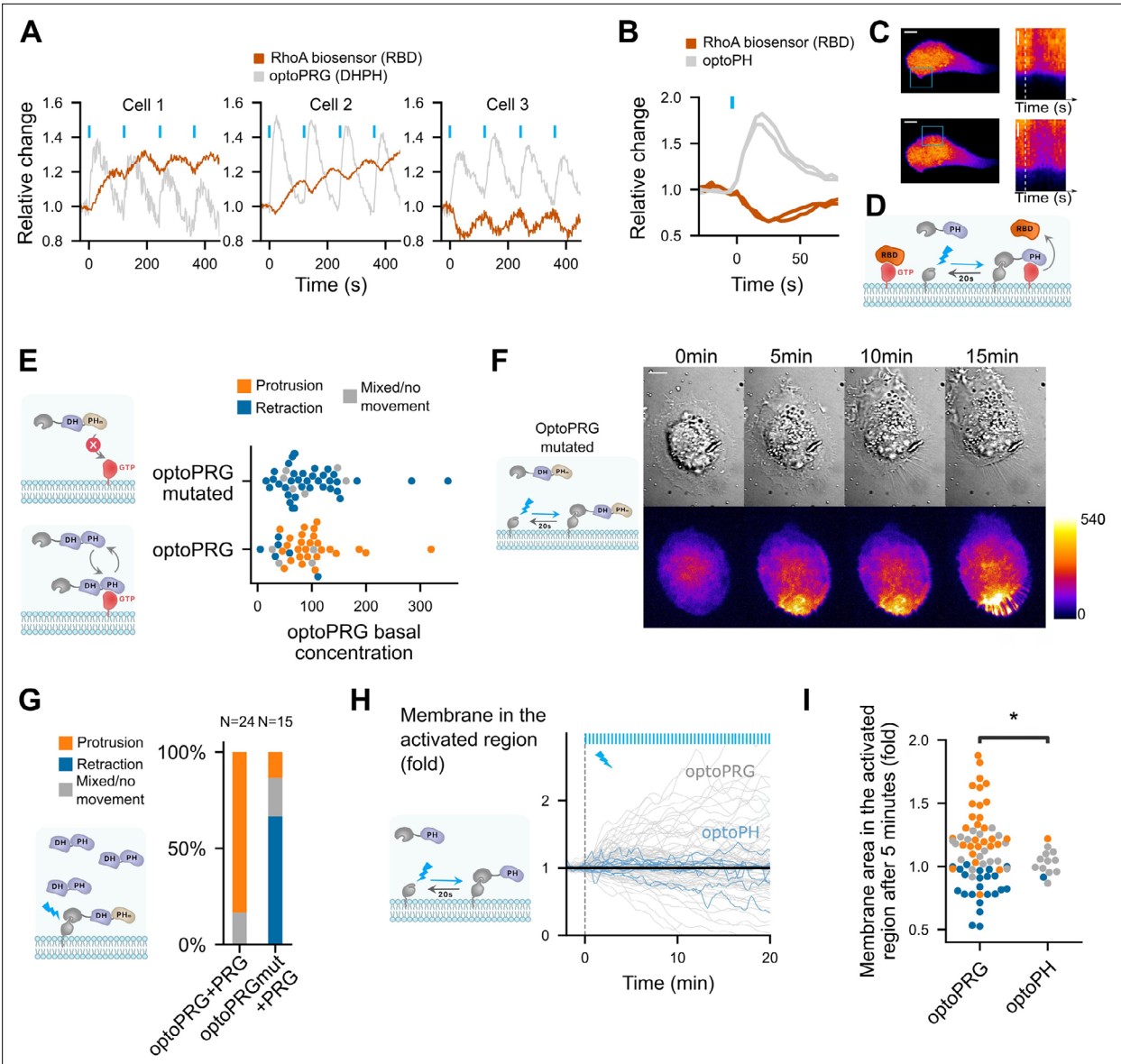

**Figure 4.** PH domain of PRG is triggering inhibition of RhoA at high PRG concentration and is necessary but not sufficient for protruding phenotype. (**A**) Three representative cells that show very different responses to optoPRG pulsatile activation. Cell 1 has a low optoPRG expression, while cells 2 and 3 have high concentration of optoPRG. Intensities are normalized by the mean intensity before the first activation ($t = 0$). (**B**) Quantification of the relative change in RhoA biosensor (RBD) after one pulse of optogenetic recruitment of the PH domain of PRG. In gray, optoPH recruitment, in red, RhoA biosensor. Light pulses are shown with blue bars. (**C**) Image of the corresponding cell. On the right, kymograph taken within the activation region. White dotted line shows the starting point of the activation. Scale bar: 10 μm. Region of activation is shown is the blue rectangle. (**D**) Scheme of the probable mechanism of PH domain dominant negative effect on RhoA. (**E**) Phenotype after optogenetic activation for optoPRG (bottom) and optoPRG with the PH mutated for no binding to RhoA-GTP (top). On the left, schemes of the expected behavior of the corresponding proteins. (**F**) Representative image of a cell transfected with the optoPRG with mutated PH, doing a retraction despite the high optoPRG concentration. (**G**) Quantification of protruding and retracting phenotypes in cells highly overexpressing non-recruitable PRG, comparing mutated and non-mutated optoPRG, with a scheme of the experiment on the left. See *Figure 4—figure supplement 1B, C* for the selected cells. (**H**) Membrane displacement of optoPH cells (in blue) with overexpressed DH–PH domain of PRG, compared to optoPRG cells (in gray). No specific protrusion can be seen. (**I**) Normalized membrane area after 5 min in the activated for optoPRG and optoPH cells. Orange: protruding cells, gray: mixed phenotype or no movement, blue: retracting cells. *$p < 0.05$ (Levene test to compare variance).

The online version of this article includes the following figure supplement(s) for figure 4:

**Figure supplement 1.** Membrane recruitment of the PH domain lowers active RhoA but is not sufficient to induce protrusion.

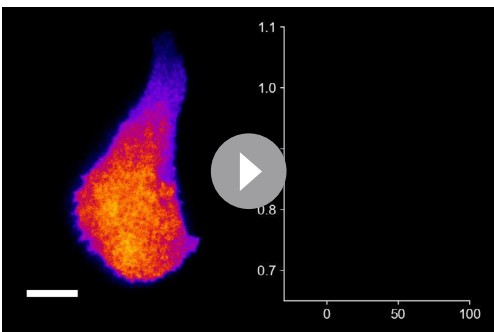

**Video 4.** RhoA biosensor dynamics upon two consecutive optoPH activation (white box). The corresponding signal quantification in the activation region is shown on the right. Scale bar: 10 μm.
https://elifesciences.org/articles/93180/figures#video4

high optoPRG levels, thus being responsible for its functional inhibition. To test our hypothesis, we designed a new optogenetic tool to recruit only the PH domain of PRG (called optoPH), while looking at the basal RhoA activity. In some cells, we could see a very clear immediate decrease of the biosensor intensity following optoPH recruitment, very similar in terms of dynamics to the decrease observed in highly expressing cells (*Figure 4B–D* and *Video 4*). This decrease was significant on the mean but not clearly visible in all cells (*Figure 4—figure supplement 1*), which was expected since the PH inhibition should depend strongly on the basal RhoA activity that might be mild in resting cells. Our result confirmed that the PH domain of PRG alone can inhibit RhoA downstream signaling, probably through a direct competitive binding as proposed in *Chen et al., 2010* (*Figure 4D*).

Next, we wondered whether such a functional inhibition plays a role in the protruding phenotype we observe with optoPRG. To test this, we mutated our optoPRG with a double mutation (F1044A and I1046E) known to prevent binding of the PH domain of PRG to RhoA-GTP (*Chen et al., 2010*). This dual mutation had a strong effect on our optogenetic activations, restricting the phenotypes to only the retracting ones, even at high basal concentrations (*Figure 4E, F*). We concluded that the PH domain of optoPRG must bind to RhoA-GTP for the protruding phenotype to happen; either before the activation – to change the cell state – or during recruitment – to prevent RhoA activity. To discriminate between these two hypotheses, we overexpressed the DH–PH domain alone in another fluorescent channel (iRFP) and recruited the mutated PH at the membrane. If the binding to RhoA-GTP was only required to change the cell state, we would expect the same statistics than in *Figure 2D*, with a majority of protruding cells due to DH–PH overexpression. On the contrary, we observed a majority of retracting phenotype even in highly expressing cells (*Figure 4G*), showing that the PH binding to RhoA-GTP during recruitment is a key component of the protruding phenotype. Few cells with very high PRG concentration still displayed small ruffles, indicating that even if much less efficient, optoPRG with mutated PH could still trigger protrusions.

These experiments suggest a necessary role of the PH domain of optoPRG for protrusions, but is it sufficient? To check this, we looked at the phenotypic response of cells that overexpress PRG, where we only recruit the PH domain. We could not see either clear protrusion or retraction happening following PH recruitment, as shown by membrane displacement in *Figure 4H, I*. Thus, the sequestration function of the PH domain is not sufficient for triggering protrusions. This points to the activation of another effector, actively responsible for protrusion formation.

## optoPRG activates Cdc42

To find the active process involved in the protruding phenotype, we turned to the two other best-known GTPases, Rac1 and Cdc42, which are main drivers of cell protrusions. Indeed, a recent work showed that the DH–PH domain of PRG was able to bind Cdc42 and activate it, thanks to conformational change operated by Gα$_S$ (*Castillo-Kauil et al., 2020*). To look at the dynamic activity of Rac1 and Cdc42, we first used a Pak Binding Domain (PBD) fused to iRFP. PAK is known for being an effector of both Rac1 and Cdc42, but more sensitive as a biosensor of Cdc42 (*Mahlandt et al., 2023*). After optogenetic activation, we saw an increase in PBD intensity in both the protruding and retracting phenotypes. After 1 min, PBD intensity continuously increases for the protruding phenotype while it remains unchanged for the retracting phenotype (*Figure 5A–C* and *Video 5*). This suggested an immediate activation of Rac1 or Cdc42 after recruitment of optoPRG, that would be inhibited after few minutes in the case of the retracting phenotype.

We then turned to recently published biosensors that are more specific to Rac1 and Cdc42 (*Nanda et al., 2023*). It revealed that Cdc42 is the Rho-GTPase specifically activated immediately

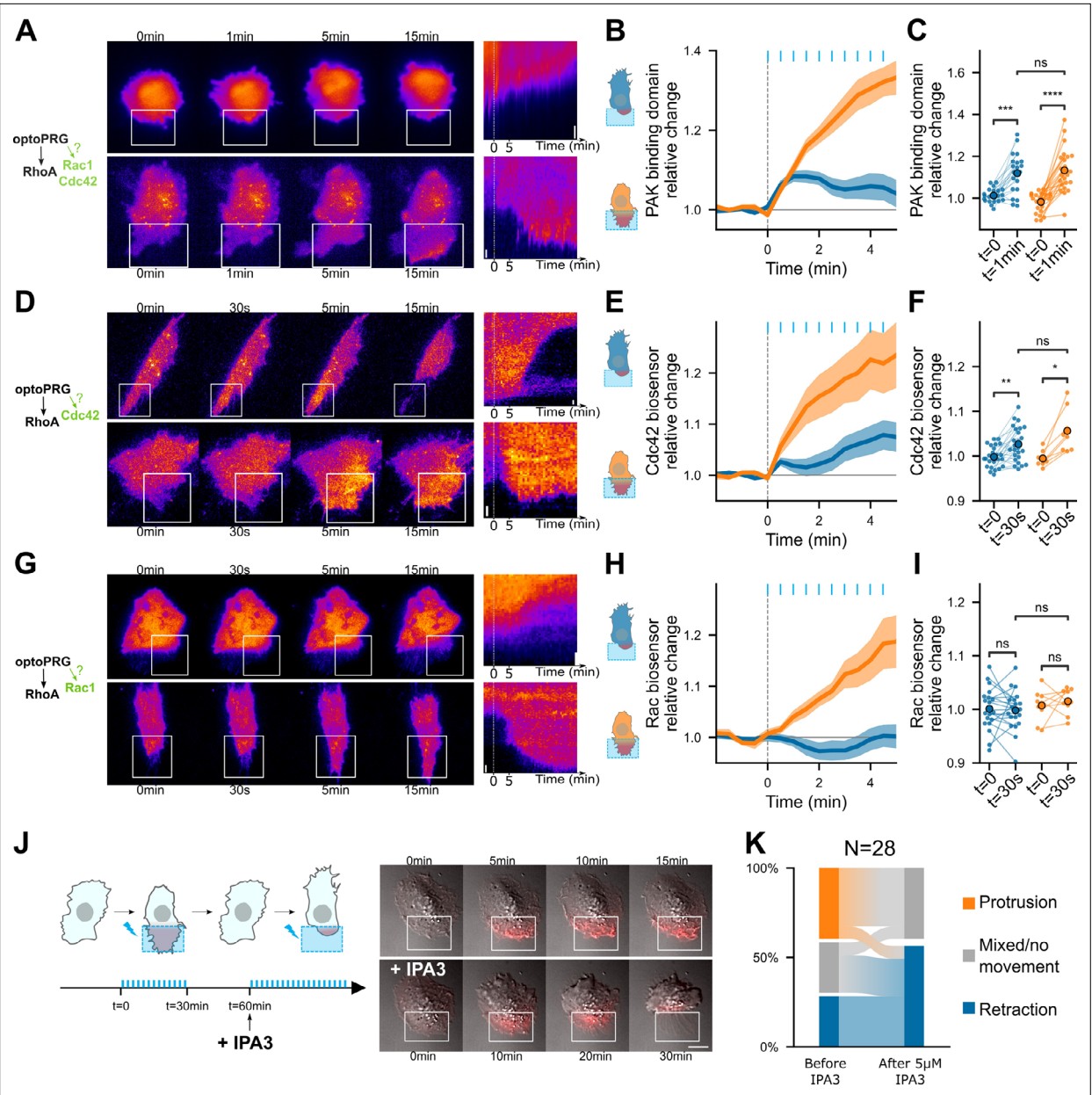

**Figure 5.** PRG activates Cdc42 and Cdc42 downstream activity is necessary for the protrusive phenotype. Representative timelapse images and kymographs of retracting (top) and protruding (bottom) cells labeled with PBD-iRFP (**A**), delCMV-mCherry-WaspGBD (**D**), and mCherry-3xp67Phox (**G**) biosensors (*Nanda et al., 2023*), activated with optoPRG starting at *t* = 0 min. White rectangles are areas of optogenetic activation. Scale bars are 10 μm. (**B, E ,H**) Corresponding mean normalized intensities are plotted against time (mean ± s.e.m.), blue for retracting cells and orange for protruding one. (**C, F, I**) Corresponding pairwise comparison for each cell of the signal inside the region of activation between the initial time and 60 s (PBD biosensor) or 30 s (Cdc42 and Rac1 biosensors). Data are grouped by phenotype. *p < 0.05, **p < 0.01, ***p < 0.001, ****p < 0.0001 (Wilcoxon test to compare *t* = 0 and *t* > 0, independent *t*-test otherwise). (**J**) Left, scheme describing the IP3 experiment. Blue bars represent optogenetic pulses (every 30 s). Half an hour after the first experiment, IPA3 is added at 5 μM. Right, representative cell showing a protruding phenotype with ruffles (top), and a retracting phenotype after addition or IPA3 (bottom). (**K**) Quantification of phenotype switches.

after optogenetic activation (*Figure 5D–F* and *Video 5*), Rac1 being activated afterwards and only in the protruding phenotype (*Figure 5G–I* and *Video 5*), most probably due to the positive feedback between Cdc42 and Rac1 (*de Beco et al., 2018*). It also confirmed that Cdc42 was activated in both phenotypes just after the optogenetic recruitment but kept being activated only in the case of the protruding phenotype.

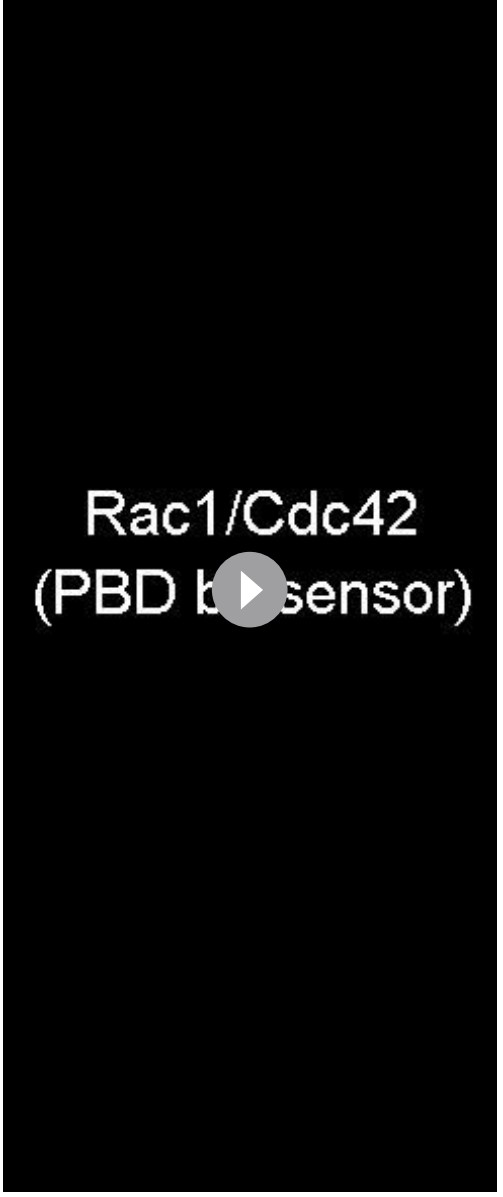

**Video 5.** PBD (Rac1/Cdc42), Cdc42, and Rac1 biosensor dynamics upon optoLARG activation (white box). Scale bar: 10 μm.

https://elifesciences.org/articles/93180/figures#video5

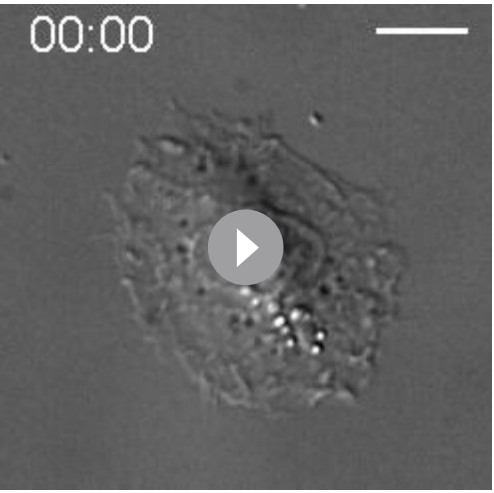

**Video 6.** Example of a cell protruding upon optoLARG activation (white box), and then retracting upon optoLARG activation (white box) after incubation with 5 μM of IPA. Differential interference contrast (DIC) transmitted light overlayed with TIRFM optoLARG signal in red. Scale bar: 10 μm.

https://elifesciences.org/articles/93180/figures#video6

To confirm that Cdc42 activation was necessary for the protrusion, we performed an experiment with IPA3, a drug targeting PAK, one of the direct effectors of Cdc42. We did a first optogenetic activation with a set of cells to know their phenotypes, then incubated our sample for 5 min with 5 μM IPA3 and activated the cells again (*Figure 5J*). A lot of cells became round, and none of them were able to protrude again upon optogenetic activation, while retracting ones were often still able to retract (*Figure 5K*). Some previously protruding cells were even able to change phenotype and retract after drug incubation (*Figure 5J, K* and *Video 6*). This confirmed us that activation of PAK through Cdc42 was required for the protruding phenotype to happen, but not for the retracting one.

## An effective model recapitulates RhoA activity dynamics and enables a control of both phenotypes in the same cell

Given the complexity of all interactions happening and the quantitative nature of our findings, we sought a minimal model that would capture RhoA biosensor dynamics and the phenotype switch as a function of optoPRG basal concentration. We also wanted to see if we could play with the quantitative properties of the light stimulation to control both phenotypes in the same cell.

To model RhoA dynamics within a region of the cell, we considered a simple reaction scheme (*Figure 6A*) where inactive RhoA is activated by the GEF (optoPRG) following mass action kinetics. The GEF can be either free or bound to active RhoA thanks to its PH domain. This complex between RhoA-GTP and the GEF, noted $GR$, is inhibiting RhoA downstream activity by titrating active RhoA but does not prevent GEF activity, as shown in *Chen et al., 2010*. The formation of the complex is characterized by the dissociation constant $K_b$. Active RhoA is assumed to be deactivated by endogenous

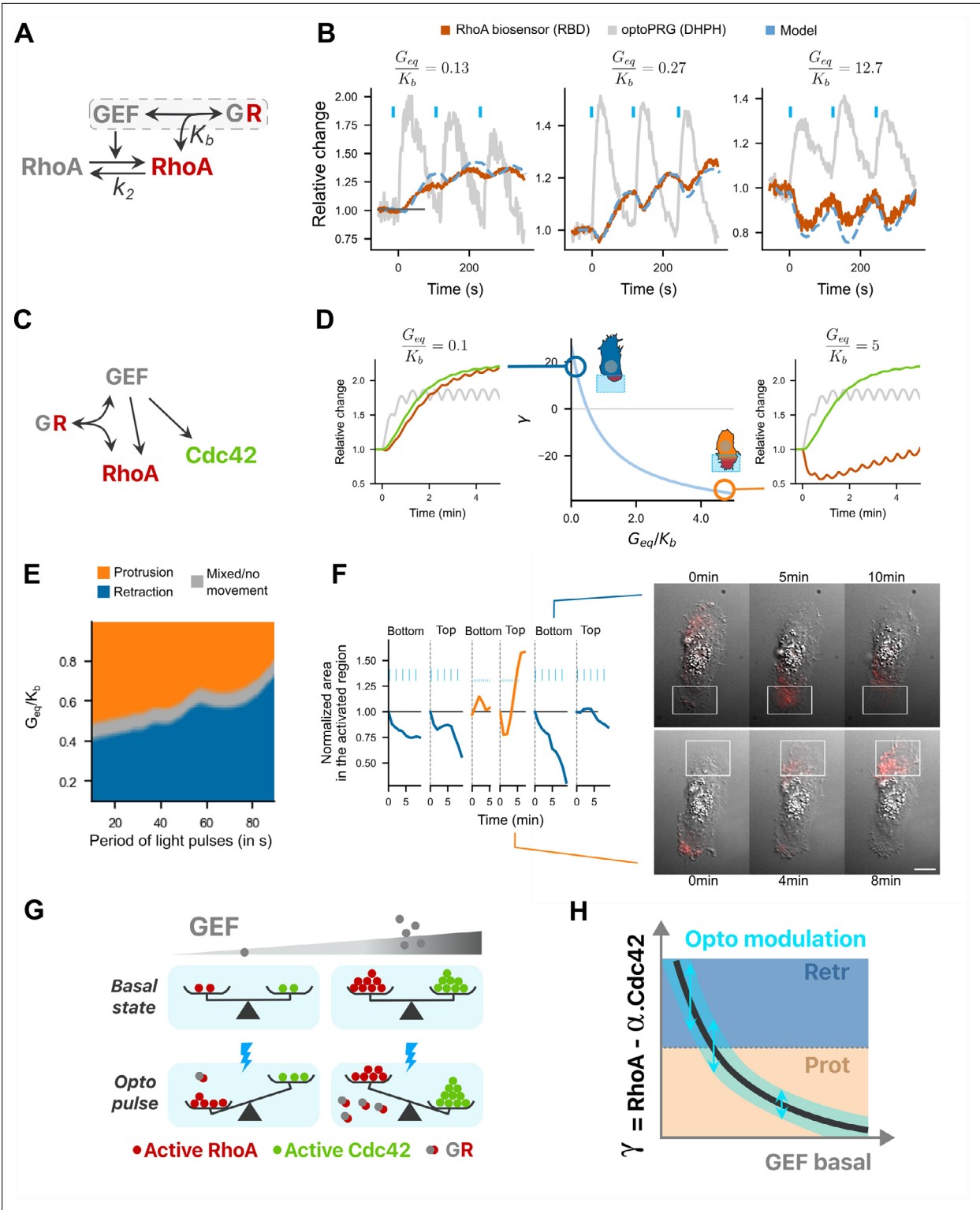

**Figure 6.** A minimal model recapitulates RhoA activity dynamics and the phenotypic switch. (**A**) Model for active RhoA dynamics. Interactions are represented with arrows, with the two main parameters of the model. (**B**) The three different RhoA dynamics are well fitted with one single free parameter, $G_{eq}/K_b$. Dotted blue line: fitted curve, with gray line $g(t)$ taken as input (optoPRG recruitment). Red line: RBD biosensor. (**C**) Complete model, adding Cdc42 to (**A**): the GEF PRG can activate both RhoA and Cdc42, but can also inhibit RhoA by directly binding to it. (**D**) Center, evolution of $\gamma$ describing the phenotype (positive for retraction and negative for protrusion) against the free parameter $G_{eq}/K_b$. Two representative dynamics are shown on the right and on the left for the same input $g(t)$, for a low and high $G_{eq}/K_b$. In gray, optoPRG recruitment to the membrane, in green,

*Figure 6 continued on next page*

*Figure 6 continued*

Cdc42 activity, in red, RhoA activity. (**E**) Map of the phenotype as function of the free parameter $G_{eq}/K_b$ and of the duration time between two pulses. (**F**) One example of two phenotypes controlled in the same cell. On the left, first 10 min of the cell area in the illuminated region for different frequencies and intensities of activation (low frequency high power every 30 s, high frequency low power every 15 s). On the right, two representative timelapse of retraction (top) and protrusion (bottom), activation is shown with the white rectangle. Scale bar: 10 µm. (**G, H**) Graphical conclusion on the model. (**G**) Balance between RhoA and Cdc42 activity is represented in function of GEF basal concentration (gray gradient), both at the basal state (top) and after optogenetic activation (bottom, with blue lightning). At low concentrations RhoA takes over. At high concentration, optoPRG binds to active RhoA and inhibits it (complex *GR*), which enables Cdc42 to take over. (**H**) Curve showing the difference between RhoA and Cdc42 activity as a function of the basal intensity of the GEF. Phenotypes are marked with the colors (blue, retraction and orange, protrusion). Optogenetic modulation happens on vertical line, with the blue range, which limits the possibility of switching from one phenotype to the other.

The online version of this article includes the following figure supplement(s) for figure 6:

**Figure supplement 1.** Example of RhoA biosensor dynamics not captured by the model and independence of the phenotype with regards to the intensity of the activating light.

GAP at a rate $k_2$, which sets the characteristic delay between GEF and RhoA-GTP dynamics, putting aside the formation of the complex *GR*. During an optogenetic activation, we assumed that the amount of the GEF, called $G_{tot}$, is changing because of local concentration increase by membrane recruitment. Introducing the dimensionless variables $r = R/R_{eq}$ and $g = G_{tot}/G_{eq}$, $G_{eq}$ and $R_{eq}$ being the values at equilibrium, and making a quasi-steady-state approximation (see Appendix 1 for a detailed derivation of the model) we obtain the following main equation for the relative evolution of actively signaling RhoA:

$$\frac{dr}{dt} = \frac{1}{1 + \frac{G_{eq}}{K_b} \cdot g} \left( k_2 \left( g - r \right) - \frac{G_{eq}}{K_b} \cdot r \frac{dg}{dt} \right).$$

This equation, which predicts the temporal evolution of active and free RhoA, $r$, for any given time-dependent GEF curve, $g$, can be solved numerically and depends on only two variables, $k_2$ and $G_{eq}/K_b$. The first variable $k_2$ can be independently estimated from optogenetic experiments with low amounts of optoPRG, for which the formation of the complex is negligible. By first estimating the kinetics of the RBD biosensor that binds RhoA-GTP ($k_{off} = 0.08 \pm 0.4 \text{s}^{-1}$), we found that $k_2 = 0.014 \pm 0.003 \text{s}^{-1}$ (Appendix 1). We are left with only one free parameter, $G_{eq}/K_b$, which characterizes the basal level of expression of the optoPRG with respect to the typical concentration at which the complex *GR* forms. This parameter changes from cell to cell, depending on the transfection efficiency. Remarkably, we could reproduce the whole family of RhoA dynamics shown in *Figure 4A* adjusting this single parameter (*Figure 6B*), even if not all experimentally observed curves (*Figure 6—figure supplement 1*), probably because our model lacks an auto-amplification process. Despite its limitation and the fact that we assumed identical cell composition (no change of gene expression apart from PRG basal levels), we can consider the model describing RhoA response to optoPRG recruitment as a good representation of biochemical reactions happening in the cell at both low and high concentrations of optoPRG.

To model the phenotype, we next added the activation of Cdc42 by the GEF (*Figure 6C*). To keep our model as simple as possible, we assumed that the deactivation rate of Cdc42 was equal to $k_2$. We further hypothesized that the binary phenotype observed at the level of membrane dynamics was the outcome of a competition between the retraction protein network triggered by RhoA (involving ROCK, myosin, mDia, and bundled actin) and the protrusion protein network triggered by Cdc42 (involving PAK, Rac, ARP2/3, and branched actin), both of which may involve numerous feedbacks and crosstalks among effectors and regulators on a timescale of few minutes. We modeled this competition by a single number $\gamma$, which computes the relative difference between integrated activities of RhoA and Cdc42 during the two first minutes after activation – approximately the time at which the phenotype is experimentally seen. As we have access to relative amounts only, we arbitrarily set the relative contribution of RhoA versus Cdc42 such that when $\gamma$ is positive the cell retracts, and when negative the cell protrudes. We added a gray zone around $\gamma=0$ (or equivalently when $G_{eq}/K_b 0.5$) to take into account the mixed phenotypes. *Figure 6D* shows the dependence of $\gamma$ as a function of $G_{eq}/K_b$. The curve is monotonically decreasing: at low $G_{eq}/K_b$ RhoA and Cdc42 dynamics are almost identical but since the cell retracts we assumed that RhoA dominates Cdc42; at high $G_{eq}/K_b$ RhoA

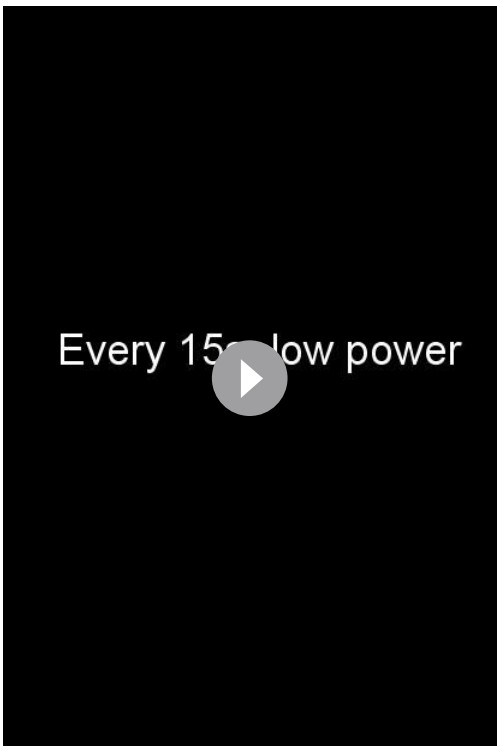

**Video 7.** Example of a cell showing the two opposite phenotypes. The cell first protrude and then contract. Differential interference contrast (DIC) transmitted light overlayed with TIRFM optoLARG signal in red. Scale bar: 10 µm.

https://elifesciences.org/articles/93180/figures#video7

is transiently inhibited by the formation of the complex and Cdc42 can dominate leading to a protrusion (see Appendix 1).

Having in hands an effective model of the phenotype, we could then explore all possible temporal patterns of activation that correspond to a family of functions $g(t)$. Given the experimental constraints of our optogenetic tool, we focused on the frequency and duration of the light pulses while taking the observed values for *on* and *off* dynamics of the iLID-SspB recruitment. We first looked at the impact of the duration of the pulse, which influences the fold increase of the function $g$ after one pulse of activation. Experimentally, we could go from a fold increase of 1.1 (to be measurable) up to 3. In our model, no cell was able to switch from retracting to protruding or vice versa by only changing the intensity of the optogenetic pulse (*Figure 6—figure supplement 1B*). This can be understood by the linearity of the model in which all terms are scaling proportionally to the intensity of the input. Then, we looked at the influence of the frequency of the light pulses on the phenotype. Interestingly, for intermediate values of $G_{eq}/K_b \sim 0.5$, we found that high frequencies lead to protruding phenotypes while low frequencies lead to retracting ones (*Figure 6E*). This results from the difference in reaction dynamics: sequestration of RhoA by the PH happens almost instantaneously, while activation of RhoA comes with a delay. This result suggests that one could switch the phenotype in a single cell by selecting it for an intermediate expression level of the optoPRG. To verify this theoretical prediction, we automatically screened cells that expressed optoPRG at a level corresponding to the transition from one phenotype to the other (14–35 a.u., see *Figure 2A*). While most cells showed mixed phenotypes irrespectively of the activation pattern, in very few cells (3 out of 90) we were able to alternate the phenotype between retraction and protrusion several times at different places of the cell by changing the frequency while keeping the same total integrated intensity (*Figure 6F* and *Video 7*).

Our model can be summarized by the following picture (*Figure 6G*). At low concentration of the GEF, both RhoA and Cdc42 are activated by optogenetic recruitment of optoPRG, but RhoA takes over. At high GEF concentration, recruitment of optoPRG lead to both activation of Cdc42 and functional inhibition of already activated RhoA, which pushes the balance toward Cdc42. In the end (*Figure 6H*), optogenetic modulation of PRG can be seen as moving the balance between RhoA and Cdc42 depending on the basal state. This means that only the cells in intermediate concentration range can use the same protein to control both antagonist responses.

## Discussion

Using the DH–PH domain of PRG, we have shown that its local recruitment to the plasma membrane can result in two opposite phenotypes: a protrusion in the activation region when highly expressed, or a retraction in the activation region when expressed at low concentrations. The known ability of the DH–PH domain to activate RhoA was confirmed in the case of retraction phenotypes. However, in protruding phenotypes, it negatively regulated RhoA activity in the first minutes after optogenetic perturbations and simultaneously activated Cdc42. These findings were summarized in a simple model that recapitulated the various experimental results. It predicted that cells with intermediate

concentrations of the optogenetic actuator could show both phenotypes depending on the frequency of the light pulses. We verified experimentally that it was the case, confirming that we had captured the main features of the underlying biochemical network.

The ability of PRG to induce the two phenotypes is supported by previously reported direct interactions. First and most obviously, the activation of RhoA by the he DH–PH domain of PRG has been well documented, the structure of the complex is even known (*Derewenda et al., 2004*). The ability of PRG DH–PH to greatly enhance RhoA-GDP switch to RhoA-GTP has been studied in vitro with purified proteins (*Gasmi-Seabrook et al., 2010*). This enhancement is assumed to come from the positive feedback loop by which RhoA in its active GTP form recruits the GEF through its interaction with the PH domain (*Chen et al., 2010*), leading to more activation of RhoA. However, this feedback loop can turn into a negative regulation for high level of GEF: the direct interaction between the PH domain and RhoA-GTP prevents RhoA-GTP binding to effectors through a competition for the same binding site. Along this line, it was shown that overexpressing the PH domain alone reduced RhoA activity (*Chen et al., 2010*). Second, the direct interaction of the DH–PH domain of PRG with Cdc42-GDP and its ability to enhance the switch to Cdc42-GTP has been shown in a previous work (*Castillo-Kauil et al., 2020*). Notably, the linker region between the DH and the PH domain is required for effective interaction, as well as $G_{\alpha s}$ activity.

Our observation of the double phenotype appeared to be a relatively general feature since we obtained it in another cell line and with at least one other GEF of RhoA. However, the protruding phenotype happened rarely with the DH–PH domain of GEF-H1, and we could not observe it at all with the DH–PH of LARG, another GEF of RhoA from the Dbl family. Thus, the precise biochemical nature of GEF domains is of importance. The interactions of the PH domain of GEF-H1 and LARG with RhoA-GTP have been described in *Medina et al., 2013*: GEF-H1 PH domain has almost the same competitive inhibition ability as PRG PH domain, while LARG PH domain is less efficient. Moreover, it seems that LARG DH–PH domain has no effect on Cdc42 (*Castillo-Kauil et al., 2020*), while nothing is known for GEF-H1 DH–PH domain. Consequently, the fact that PRG DH–PH can trigger protrusions reproducibly compared to other GEFs could be explained by its ability to efficiently sequester RhoA-GTP while activating Cdc42 at the same time. These two combined properties would allow the GEF to be expressed at a high basal level required for the protruding phenotype since the sequestration by the PH domain prevents RhoA overactivation and Cdc42 activation prevents cell rounding through its competition with RhoA. This could also explain the increase in cell size for protruding cells. Supporting this hypothesis, we observed that expressing transiently optoGEF-H1 and optoLARG was much harder than for optoPRG, many cells becoming round or dying when positively transfected. This underlines the attention that needs to be paid to the choice of specific GEF domains when using optogenetic tools. Tools using DH–PH domains of PRG have been widely used, both in mammalian cells and in *Drosophila* (with the orthologous gene RhoGEF2), and have been shown to be toxic in some contexts in vivo (*Rich et al., 2020*). Our study confirms the complex behavior of this domain which cannot be reduced to a simple RhoA activator.

Interestingly, PRG is known for its role in cell migration, both at the rear (*Iwanicki et al., 2008*) and at the front (*Nanda et al., 2023*). Given that FRET measurements that are sensing GEFs activity reports RhoA activity at the protruding front (*Pertz et al., 2006*) and biosensors of RhoA-GTP reports activity at the retracting back (*Mahlandt et al., 2021*), our results might solve this paradox: PRG would activate Cdc42 at the front meanwhile activating RhoA at the back. PRG is also known for being prometastatic and is overexpressed in different cancers favoring migration and epithelial to mesenchymal transition (*Struckhoff et al., 2013*; *Du et al., 2020*; *Ding et al., 2018*). In these pathological cases, PRG was studied as a promoter of RhoA activity. However, our results point clearly toward a possible switch in PRG role when overexpressed, acting more on Cdc42 activity. Such a switch in function could be a mechanism happening for other GEFs or proteins and should be considered when designing therapies.

We demonstrated that changing the dynamics of one single protein is enough to revert its function when being recruited to the membrane. Even if the context was quite specific (cells with a specific concentration), such multiplexing may be happening in vivo, where dynamics and local concentrations can highly vary in the cytoplasm and in different subcellular domains. We were limited here by the dynamic of the optogenetic dimer, but our model suggests that optogenetic tools with a shorter lifetime than iLID/SSPB (~20 s) would ease the phenotypic switch in the same cell. Similarly, kinetics of

endogenous interactions might be tuned on fast timescales by natural molecular circuitries, allowing cells to multiplex signals by this means. The fact that a protein can have different functions based on its dynamics is not new (*Purvis and Lahav, 2013*). However, examples that demonstrate a causal relationship between activation of a protein and opposite cellular responses exist mostly at the transcriptional level, on the timescale of hours (*Toettcher et al., 2013*). To our knowledge, there are only two examples, both involving optogenetics, of such a dual response on shorter timescales and for protein–protein interactions. In the first example, it was shown that two different acto-adhesive structures could form in response to either Src recruitment or clustering (*Kerjouan et al., 2021*); the specificity being encoded here by the dynamics of Src nanoclusters at the adhesive sites. In the second example, RhoA activation by uncaging of a GEF of RhoA triggered focal adhesion growth via Src activation only at submaximal levels, revealing a selection of cellular response by signal amplitude (*Ju et al., 2022*).

Altogether, we have revealed and explained a striking example of protein multiplexing, while underscoring the crucial role of protein dynamics for signal transduction. We have also raised open questions about the link between signaling proteins and their functions, particularly in contexts where they are highly overexpressed, as often observed in cancer.

## Materials and methods

### Key resources table

| Reagent type (species) or resource | Designation | Source or reference | Identifiers | Additional information |
|---|---|---|---|---|
| Cell line (*Homo sapiens*) | hTERT RPE1 (immortalized, normal, female) | ATCC | ATCC Cat# CRL-4000, RRID:CVCL_4388 | |
| Cell line (*H. sapiens*) | HeLa (adenocarcinoma, female) | ATCC | ATCC Cat# CCL-2, RRID:CVCL_0030 | |
| Transfected construct (*H. sapiens*) | pLVX: MRLC-iRFP | Coppey lab | | Plasmid to label myosin regulatory light chain |
| Transfected construct (*H. sapiens*) | pLVX: Lifeact-iRFP | Coppey lab | | Plasmid to label polymerizing actin |
| Transfected construct (*H. sapiens*) | pLVX: PBD-iRFP | Coppey lab | | Plasmid for a biosensor of Rac and Cdc42 activity |
| Transfected construct (*H. sapiens*) | pCMV:PRG(DHPH)-RFPt-SspB-P2A-mVenus-iLID-CAAX | Casano lab | | Plasmid for an optogenetic construct recruiting the DH–PH of PRG to the membrane |
| Transfected construct (*H. sapiens*) | pLL7:PRG(DHPH)-iRFP-SspB-P2A-mVenus-iLID-CAAX | Coppey lab | | Plasmid for an optogenetic construct recruiting the DH–PH of PRG to the membrane |
| Transfected construct (*H. sapiens*) | pLL7:VenusiLID-CAAX. pCMV:PRG(PH)-iRFP-SspB-P2A-mVenus-iLID-CAAX | Coppey lab | | Plasmid for an optogenetic construct recruiting the PH of PRG to the membrane |
| Transfected construct (*H. sapiens*) | pCMV:PRG(PH)-RFPt-SspB-P2A-mVenus-iLID-CAAX | Coppey lab | | Plasmid for an optogenetic construct recruiting the PH of PRG to the membrane |
| Transfected construct (*H. sapiens*) | delCMV-mCherry-3xp67Phox | Dehmelt lab | | Plasmid for a biosensor of Rac activity |
| Transfected construct (*H. sapiens*) | delCMV-mCherry-WaspGBD | Dehmeltlab | | Plasmid for a biosensor of Cdc42 activity |
| Chemical compound, drug | IPA-3 | Calbiochem | CAS 42521-82-4 | 5 µM |
| Software, algorithm | Python | Python Software Foundation | RRID:SCR_008394 | |
| Software, algorithm | Fiji, ImageJ | | RRID:SCR_002285 | |
| Software, algorithm | MetaMorph | Molecular Devices | RRID:SCR_002368 | |

## Cell culture

hTERT RPE1 cells (CRL-4000 strain, ATCC, Manassas, VA) were cultured at 37°C with 5% $CO_2$ in Dulbecco's modified Eagle's/F-12 medium supplemented with 10% fetal bovine serum, GlutaMAX (2 mM) and penicillin (100 U/ml)–streptomycin (0.1 mg/ml). Cells were passaged twice a week in a ratio of 1/10 by washing them with PBS (1×) solution and dissociating using TrypLE Express (Thermo Fisher Scientific, Waltham, MA) reagent for 3–5 min.

## Plasmids

dTomato-2xrGBD (Plasmid #129625) (RhoA biosensor), pLL7:Venus-iLID-CAAX (#60411), and 2XPDZ-mCherry-Larg(DH) (Plasmid #80407) plasmids were bought from Addgene (Watertown, MA).

pLVX: MRLC-iRFP, pLVX: Lifeact-iRFP pLVX: PBD-iRFP (PAK biosensor) plasmids were made by Simon de Beco (Institut Curie, France). pCMV:PRG(DHPH)-RFPt-SspB-P2A-mVenus-iLID-CAAX is a gift from Alessandra Casano (EMBL). pLL7:PRG(DHPH)-iRFP-SspB-P2A-mVenus-iLID-CAAX was subcloned by Maud Bongaerts (Institut Curie, France) from PRG(DHPH)-CRY2-mCherry and pLL7:VenusiLID-CAAX. pCMV:PRG(PH)-iRFP-SspB-P2A-mVenus-iLID-CAAX was subcloned by Benoit Boulevard (Institut Curie, France) from pCMV:PRG(DHPH)-RFPt-SspB-P2A-mVenusiLID-CAAX. pCMV:PRG(PH)-RFPt-SspB-P2A-mVenus-iLID-CAAX was designed in the lab and synthesized by Twist Bioscience (Twist Bioscience, San Francisco). Rac1 and Cdc42 biosensors, delCMV-mCherry-3xp67Phox and delCMV-mCherry-WaspGBD, respectively, described in *Nanda et al., 2023*, were gifts from Leif Dehmelt (Dortmund and Max Planck Institute of Molecular Physiology).

## Transfection

Transfections were performed using jetPRIMEversatile DNA/siRNA transfection reagent according to the manufacturer's protocol. Different ratio of plasmid DNA was used depending on the constructs, and a ratio of 2:1 of transfection reagent and DNA. Experiments were performed at least 30 hr after DNA transfection, and 48 hr after siRNA transfection.

## Drug assay

For IPA-3 (p21-Activated Kinas Inhibitor III, CAS 42521-82-4, Calbiochem), between 20 and 30 cells were first selected and optogenetically activated for 30 min. Thirty minutes after the end of the first activation, the medium was replaced by the drug diluted in complete DMEM/F-12 medium at specified concentrations, and optogenetic activation was done again 5 min after medium replacement.

## Imaging

Imaging was performed at 37°C in 5% $CO_2$. Two different microscopes have been used: an IX83 and an IX71 (Olympus, Melville, NY) both with inverted fluorescence and differential interference contrast (DIC) and controlled with MetaMorph software (Molecular Devices, Eugene, OR). Both microscopes were equipped with a 60× objective (NA = 1.45), motorized stage and filter wheel with SmartShutter Lambda 10-3 control system (Sutter Instrument Company, Novato, CA), a stage-top incubation chamber with temperature and $CO_2$ control (Pecon, Meyer Instruments, Houston, TX), a laser control system with azimuthal TIRF configuration (iLas2, Roper Scientific,Tucson, AZ). The IX71 was equipped with an ORCA-Flash5.0 V3 Digital CMOS camera (Hamamatsu Photonics K.K., Japan), a *z*-axis guiding piezo motor (PI, Karlsruhe, Germany), a CRISP autofocus system (ASI, Eugene, OR), and a DMD pattern projection device (DLP Light Crafter, Texas instruments, Dalas, TX), illuminated with a SPECTRA Light Engine (Lumencor, Beaverton, OR) at 440 ± 10 nm. The IX83 has a built-in z-piezo and autofocus, and was equipped with an Evolve EMCCD camera (Photometrics, Tucson, AZ) and an FRAP configuration (iLas2, Roper Scientific, Tucson, AZ).

## Local optogenetic illumination

Local illumination was performed thanks to a DMD placed on the optical path. DMD used in the experiments was a DLP4500 with a LC4500 controller from Keynote Photonics (Keynotes Photonics, Allen, TX). The chip has a dimension of 1140x912 micromirrors (6161.4 µm × 9855 µm), able to generate 8-bit grayscale patterns. We generated custom illumination patterns using the DMD and a blue LED illumination source SPECTRA Light Engine (Lumencor, Beaverton, OR).

## Cell finder

To find transfected cells on coverslips with a wide range of fluorescence intensities, working with single cells, we developed a small Python software (Python Software Foundation. Python Language Reference, version 3.9.5.), which we named Cell finder, that greatly facilitates the search for transfected cells. This software, available on Github https://github.com/jdeseze/cellfinder (copy archived at *jdeseze, 2023*) scans the entire available area, finds any fluorescent object larger than a predefined size (with a threshold to define what is fluorescent), and produces the resulting list of locations in customized format for the Metamorph imaging software (the version on Github has an option to be used with Micromanager). If the number of cells found is too important, a Python-based GUI interface is used to select only the desired positions based on the image acquired during the search trajectory. It allows seeding cells at low density to scatter them, and still have dozens of transfected cells within one 25-mm coverslip for the experiments.

## Image analysis

All image analyses have been done with homemade script, using Python (Python Software Foundation. Python Language Reference, version 3.9.5) through the napari interface (*Chiu and Clack, 2022*). All movies are created with Fiji, as well as picture montages. The kymographs have been done thanks to the *Reslice* function in Fiji, with different linewidths depending on the with of the cell. Quantification have been done thanks to custom plugins for napari imaging software (*Chiu and Clack, 2022*) developed in the lab and available on github (https://github.com/jdeseze/napari-intensity-measurements). Segmentation used optical flow Farneback algorithm in a similar way than in *Robitaille et al., 2022* for membrane displacement measurement, or classical thresholding of fluorescent channel for intensity measurements.

## Data processing

### Surface displacement

Surface displacement is the area of the intersection between the activated area and cell segmentation. *Normalization*: Biosensors intensities are calculated the following way. First, background is subtracted. Second, mean intensity in the intersection between activated area and cell segmentation is calculated. Third, this intensity is divided by the mean intensity in the non-activated part of the cell. Fourth, the intensity is normalized by the intensity before the optogenetic activation. *Dotplots of intensities*: The normalized curved at the specific timepoints specified in each figure are plotted as swarmplots, with the means. *Images and movies*: For DIC images, raw image is divided by the Gaussian-filtered image with a large diameter, from 30 to 50 pixels, to correct for uneven illumination. *Sankey diagrams*: Sankey diagrams were done by modifying a python library called pySankey (https://pypi.org/project/pySankey/). *Persistence*: Persistence was calculated as the ratio between the actual distance from the initial point and the sum of the absolute value of all displacements. *Myosin ratio*: Clusters were cut it in half along the *x*-axis, and the ratio between left (activated) and right (non-activated) part was measured.

### Modeling

The modeling was performed with Python software (Python Software Foundation. Python Language Reference, version 3.9.5). The differential equations were integrated thanks to the *odeint* function from scipy.integrate package, and all the fitted parameters were found by the least squares method, using *fmin_powell* function from scipy.optimize for minimization, which uses Powell minimization's method.

### Statistical tests

We used *t*-test for comparing independent datasets, and Wilcoxon rank sum test when comparing the same cells at different timepoints. All tests were performed using Python software with the SciPy library. Statistical details of each experiment can be found in the figure legends.

## Acknowledgements

The work was done with support from the LabEx Cell(n)Scale (ANR-10-LABX-0038), Labex and Equipex IPGG (ANR-10-NANO0207), Idex Paris Science et Lettres (ANR-10-IDEX-0001-02 PSL), French National Research Infrastructure France-BioImaging (ANR-10-INBS-04), and Institut Convergences Q-life (ANR-17-CONV-0005). JDS thanks AMX from Ecole Polytechnique and the ARC foundation.

## Additional information

### Funding

| Funder | Grant reference number | Author |
|---|---|---|
| Agence Nationale de la Recherche | ANR-10-LABX-0038 | Mathieu Coppey |
| Agence Nationale de la Recherche | ANR-10-NANO0207 | Mathieu Coppey |
| Agence Nationale de la Recherche | ANR-10-IDEX-0001-02 PSL | Mathieu Coppey |
| Agence Nationale de la Recherche | ANR-10-INBS-04 | Mathieu Coppey |
| Agence Nationale de la Recherche | ANR-17-CONV-0005 | Mathieu Coppey |
| Association pour la Recherche contre le Cancer | ARC | Jean de Seze |

The funders had no role in study design, data collection, and interpretation, or the decision to submit the work for publication.

### Author contributions

Jean de Seze, Conceptualization, Data curation, Software, Formal analysis, Validation, Investigation, Visualization, Methodology, Writing – original draft, Project administration, Writing – review and editing; Maud Bongaerts, Formal analysis, Investigation, Project administration; Benoit Boulevard, Resources; Mathieu Coppey, Conceptualization, Resources, Data curation, Formal analysis, Supervision, Funding acquisition, Validation, Investigation, Visualization, Methodology, Writing – original draft, Project administration, Writing – review and editing

### Author ORCIDs

Jean de Seze https://orcid.org/0000-0002-6183-4303
Maud Bongaerts https://orcid.org/0000-0003-1971-3767
Mathieu Coppey https://orcid.org/0000-0001-8924-3233

Reviewer #1 (Public review): https://doi.org/10.7554/eLife.93180.4.sa1
Reviewer #2 (Public review): https://doi.org/10.7554/eLife.93180.4.sa2
Author response https://doi.org/10.7554/eLife.93180.4.sa3

## Additional files

### Supplementary files

Figure 1—source data 1. Code and data to produce *Figure 1*. See the README file.
Figure 2—source data 1. Code and data to produce *Figure 2*. See the README file.
Figure 3—source data 1. Code and data to produce *Figure 3*. See the README file.
Figure 4—source data 1. Code and data to produce *Figure 4*. See the README file.
Figure 5—source data 1. Code and data to produce *Figure 5*. See the README file.
Figure 6—source data 1. code and data to produce *Figure 6*. See the README file.

MDAR checklist

## Data availability

Source data files with numerical data and Python Source Code for all the graphs in the figures are provided as a zip supplementary file attached to each figure in this submission. Raw imaging data for all the figures are available in the BioImage Archive repository at https://www.ebi.ac.uk/biostudies/bioimages/studies/S-BIAD1842 with BioStudies accession number S-BIAD1842. Home made software for data analysis is available at https://github.com/jdeseze/napari-intensity-measurements (copy archived at *de Seze, 2025a*). Napari plugin for opening Metamorph images is available at https://github.com/jdeseze/napari-metamorph (copy archived at *de Seze, 2025b*). Plasmids from our lab will be shared upon request.

The following dataset was generated:

| Author(s) | Year | Dataset title | Dataset URL | Database and Identifier |
|---|---|---|---|---|
| Coppey M, de Seze J | 2025 | Optogenetic control of a GEF of RhoA uncovers a signaling switch from retraction to protrusion | https://www.ebi.ac.uk/biostudies/bioimages/studies/S-BIAD1842 | BioImage Archive, S-BIAD1842 |

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

## Appendix 1

## An effective model recapitulates reaction kinetics and enables a control of both phenotypes in the same cell

Given the complexity of all interactions happening, we sought for a minimal model that would capture the simple dependence of the phenotype with optoPRG concentration, as well as the biosensors dynamics.

Let us summarize the different facts that are essential to the model:

- Two opposite polarization phenotypes can be triggered by the recruitment of the same DH–PH domain of PRG at the membrane, in the same cell line, could potentially mean in the same biochemical environment if overexpression of PRG is not affecting gene expression.
- Phenotype can be predicted by the absolute concentration of the DH–PH domain of PRG before activation independently of the recruitment. This means that overexpression has an influence on cell state, defined as protein basal activities or concentrations before activation.
- RhoA, the direct effector of PRG, is activated in the retracting phenotype, but its activation is delayed in the protruding one.
- PH domain binding to RhoA-GTP is required for protruding phenotype but not sufficient, and it is acting as an inhibitor of RhoA activity.
- The DH–PH domain of PRG activates Cdc42, which is required for protruding phenotype.

To keep it as simple as possible, we considered that the resulting phenotype was the result of a competition between the relative amount of RhoA-GTP and Cdc42-GTP after 2 min, roughly the time when phenotypic changes start to be seen. Reactions are modeled as mass action kinetics happening at the area of activation – even for enzymatic reactions with the GEF, which is justified if the reaction is happening far from saturation and seems to be the case for GEFs of RhoA at least in another example (*Kamps et al., 2020*). We considered the optogenetic recruitment as a variation of the total amount of PRG in this area, noted $G_{tot}$. As described above, three interactions of PRG will be considered: the interaction with RhoA-GDP inducing a switch to RhoA-GTP, the interaction with Cdc42-GDP inducing a switch to Cdc42-GTP, and the interaction of the PH domain with RhoA-GTP. This last reaction gives rise to a complex (noted $GR$) that is inhibiting RhoA activity, but does not prevent GEF activity, as shown in *Chen et al., 2010*.

These assumptions lead to three following reactions:

a. $\text{RhoA}^{\text{GDP}} \underset{k_2}{\overset{K_1 \cdot G_{tot}}{\rightleftharpoons}} \text{RhoA}^{\text{GTP}}$, with typical concentration $K_a = \dfrac{k_2}{k_1}$

b. $\text{RhoA}^{\text{GDP}} + \text{G} \underset{k_4}{\overset{k_3}{\rightleftharpoons}} \text{GR}$, with typical concentration $K_b = \dfrac{k_4}{k_3}$

c. $\text{Cdc42}^{\text{GDP}} \underset{k_6}{\overset{K_5 \cdot G_{tot}}{\rightleftharpoons}} \text{Cdc42}^{\text{GTP}}$, with typical concentration $K_c = \dfrac{k_5}{k_6}$

In the following, we call $R$ and $C$ the concentrations of $\text{RhoA}^{\text{GTP}}$ and $\text{Cdc42}^{\text{GTP}}$ in their active forms, and $\bar{R}$ and $\bar{C}$ the concentrations of $\text{RhoA}^{\text{GDP}}$ and $\text{Cdc42}^{\text{GDP}}$ in their inactive forms. We can add the three conservation equations for the concentrations:

· $R_{\text{tot}} = \bar{R} + R + GR$

· $G_{\text{tot}} = G + GR$

· $C_{\text{tot}} = \bar{C} + C$

which leads to the following system of equations:

$$\begin{cases} \dfrac{dR}{dt} = k_1.G_{\text{tot}}\left(R_{\text{tot}} - R - GR\right) - k_2 R - \dfrac{dGR}{dt} \\ \dfrac{dGR}{dt} = k_3 R \left(G_{\text{tot}} - GR\right) - k_4 GR \\ \dfrac{dC}{dt} = k_5 G_{\text{tot}} \left(C_{\text{tot}} - C\right) - k_6 C \end{cases}$$

We added few hypotheses, to reduce the number of unknown parameters:

- Many GAPs are common to Cdc42 and RhoA (*Müller et al., 2020*), we considered their deactivation rates to be similar: $k_6 \approx k_2$.
- GTPases activated with strong stimuli – giving rise to strong phenotypic changes – lead to only 5% of the proteins in a GTP-state, both for RhoA and Cdc42 (*Pertz, 2010*; *Ren et al., 1999*; *Benard et al., 1999*). Therefore, we assumed that $C << C_{\text{tot}}$ and $R << R_{\text{tot}}$, even at high

concentrations of optoPRG. We also considered that a few proportions of both optoPRG and RhoA$^{GTP}$ were in complex, so that $GR << R_{tot}$ and $GR << G_{tot}$.

- As we saw that the inhibition of RhoA by the PH domain of optoPRG was faster than the activation of RhoA, we considered that the reaction (b) was constantly at equilibrium. Thus, the complex $GR$ can be expressed as a function of $G_{tot}$ and $R$ (which will both evolve in time), as the following: $GR(t) = \frac{G_{tot}(t)R(t)}{K_b}$

With these hypotheses, before optogenetic activation, we have the following equilibria:

$$\begin{cases} R_{eq} = \dfrac{G_{eq}R_{tot}}{K_a} \\ GR_{eq} = \dfrac{G_{eq}R_{eq}}{K_b} \\ C_{eq} = \dfrac{G_{eq}C_{tot}}{K_c} \end{cases}$$

with $G_{tot}(t < 0) = G_{eq}$.

## Modeling RhoA dynamics

Introducing the new variables $r = R/R_{eq}$, $c = C/C_{eq}$, $gr = GR/GR_{eq}$, and $g = G_{tot}/G_{eq}$ leads to the much simpler and independent equations:

$$\begin{cases} \dfrac{dr}{dt} = k_2(g - r) - \dfrac{GR_{eq}}{R_{eq}} \cdot \dfrac{dgr}{dt} \\ \dfrac{dc}{dt} = k_2(g - c) \end{cases}$$

Expressing $\frac{dgr}{dt}$ as a function of $r$ and $\frac{dr}{dt}$, we obtain the following equation for the evolution of $r$:

$$\frac{dr}{dt} = \frac{1}{1 + \dfrac{G_{eq} * g}{K_b}}\left(k_2(g - r) - \frac{G_{eq} * r}{K_b}\frac{dg}{dt}\right)$$

Which contains only two unknown parameters: $k_2$ and $G_{eq}/K_b$. This second parameter will be the free parameter changing from cell to cell, due to the difference in optoPRG expression.

To evaluate $k_2$, we considered the measurements made with the biosensor of RhoA activity. This biosensor has its own intrinsic dynamic associated to a given $k_{on}$ and a $k_{off}$. Considering that only a small portion is bound to RhoA$^{GTP}$ at any moment, the dynamic of the biosensor will follow the one of RhoA$^{GTP}$ with a delay, given by $k_{off}$ in the following equation: $\frac{db}{dt} = k_{off}(r - b)$ where $b$ is the relative changes of the biosensor intensity.

$k_2$ and $k_{off}$ can be estimated independently. When the PH domain is recruited alone, if we consider that the endogenous activities of GEFs and GAPs are slow compared to the binding of the PH to RhoA$^{GTP}$, we obtain $1 - rt \propto 1 - p$, where $p$ is the relative amount of PH domain. Thus, $k_{off}$ can be fitted independently of the $k_2$, and we found that $k_{off} = 0.08 \pm 0.4s^{-1}$ (**Appendix 1—figure 1**).

We can than estimate $k_2$ by looking at cells with a very low concentration of optoPRG. In these cells, we expect the formation of the complex to be negligible compared to the activation of RhoA by the PH domain. Thus, the evolution of $r$ will be given by the simple equation:

$$\frac{dr}{dt} = k_2(g - r)$$

Taking the mean value of the $k_{off}$ found above, we can fit three different dynamics in three different cells with low concentration, as shown in **Appendix 1—figure 2**, which gives $k_2 = 0.014 \pm 0.003s^{-1}$. As seen here, this very simple model seems to well represent the data that we get by inducing pulses of RhoA activity at low concentration.

As stated above, we are left with one free parameter, $G_{eq}/K_b$, for the equation describing the evolution of RhoA$^{GTP}$, which is the parameter changing from cell to cell, due the different intensities of transfection.

Taking the two mean values of $k_{off}$ and $k_2$ fitted above, we could reproduce the very different dynamics we had observed with the RBD biosensor by fitting only the ratio $G_{eq}/K_b$, as shown in

*Figure 3A* of the main manuscript. Indeed, we recovered the behavior of low transfected cells (low $G_{eq}/K_b$) where only the activation of RhoA is present, the behavior of highly transfected cells (high $G_{eq}/K_b$) where the recruitment of optoPRG leads to a inhibition of RhoA, and the behavior of cells with medium transfection, where complex dynamics with both inhibition and activation could be well fitted.

## Adding Cdc42 to model the double phenotype

Now, let us consider RhoA and Cdc42 responses during 5 min with simulated optoPRG inputs comparable to the one of the experiments. For $G_{eq}/K_b$, we choose values ranging between the extreme boundaries found by fitting the different curves previously.

As said in the main manuscript, the activity of RhoA will be in competition with the activity of Cdc42, that is also activated by optoPRG. The resulting phenotype will be a retraction if the absolute increase of RhoA$^{GTP}$ is superior to the absolute increase of Cdc42$^{GTP}$, with an unknown factor $\alpha$ that represents the efficiency of both GTPases to produce their specific phenotype. As we saw that the resulting phenotype was determined 2 min after the first activation, we choose this timepoint as the one where the other cellular processes and feedbacks take over. Thus, we compared the whole activity of RhoA and Cdc42 during the two first minutes after activation. This gives the following conditions:

$$\begin{cases} \int_0^2 (R - R_{eq}) \, dt > \alpha \int_0^2 (C - C_{eq}) \, dt \Rightarrow \text{retraction} \\ \int_0^2 (R - R_{eq}) \, dt < \alpha \int_0^2 (C - C_{eq}) \, dt \Rightarrow \text{protrusion} \end{cases}$$

We can define $\alpha' = \alpha \frac{C_{tot} K_a}{R_{tot} K_c}$ to have the equivalent condition:

$$\begin{cases} \gamma > 0 \implies \text{retraction} \\ \gamma < 0 \implies \text{protrusion} \end{cases}$$

At low concentrations, $r$ and $c$ have the same dynamic, while the phenotype is always a retracting one. Therefore, we expect $\alpha'$ to be lower than one.

For the standard type of activation, we saw that the switch between retracting and protruding phenotype was happening for $G_{eq}/K_b \approx 0.5$. Thus, we choose $\alpha' = 0.24$ so that $\gamma (G_{eq}/K_b = 0.5) = 0$. We also added a 'gray zone' around 0, where the cell would not do a mixed phenotype because activities of RhoA and Cdc42 are of the same magnitude. This gray zone will be materialized by an arbitrary value, such that $\gamma \vee 3 \implies mixed\, phenotype$.

## Modeling optogenetic pulses

To model the pulses of optoPRG recruitments, we constructed a simple model of equilibrium between iLID at the membrane and SspB in the cytosol. We considered that with one pulse of blue light, the quantity of high affinity iLID increases with a factor $f$, and then returns to equilibrium with a half-life of $\tau_{1/2} = 20s$ (*Guntas et al., 2015*). We knew that PRG has a certain affinity for the membrane, we thus added a second timescale $1/k_{off}$ to take it into account.

With this, and the hypothesis that SspB is in excess, we obtain the following equation for one pulse of activation: $\frac{dg}{dt} = k_{off} \left( 1 + (f - 1) * e^{\frac{-t}{\tau_{1/2}}} - g \right)$. This equation can be solved analytically, and $f$ and $k_{off}$ can be fitted to a standard pulse of activation, as seen in *Appendix 1—figure 3*.

We used these values of $k_{off} = 1/15 \text{s}^{-1}$ and $f = 1.5$ for all simulations shown in *Figure 6* of the main manuscript.

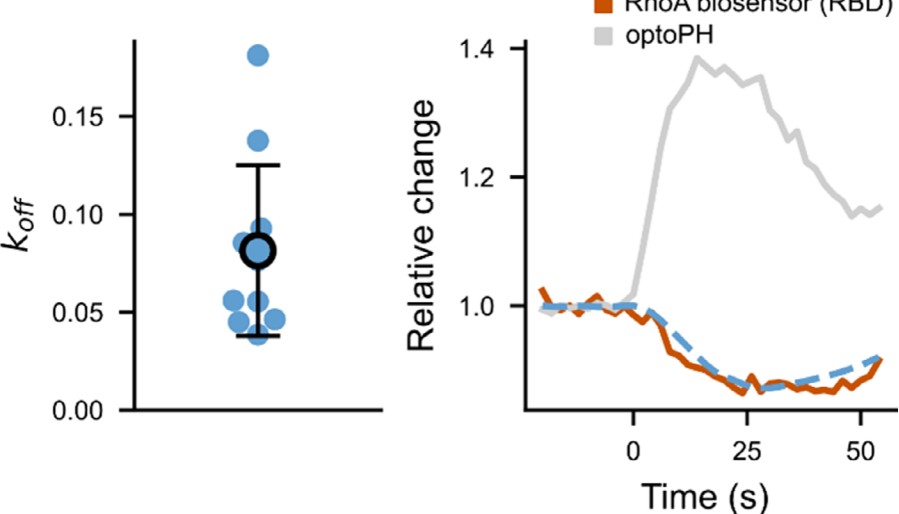

**Appendix 1—figure 1.** Fit of the biosensor $k_{\text{off}}$. On the left, $k_{\text{off}}$ values fitted from different cells ($N$ = 3 cells, with repeated measurements). On the right, an example of a fitted RBD curve (fit in blue, data in brown), plotted together with the normalized optoPH recruitment (gray curve).

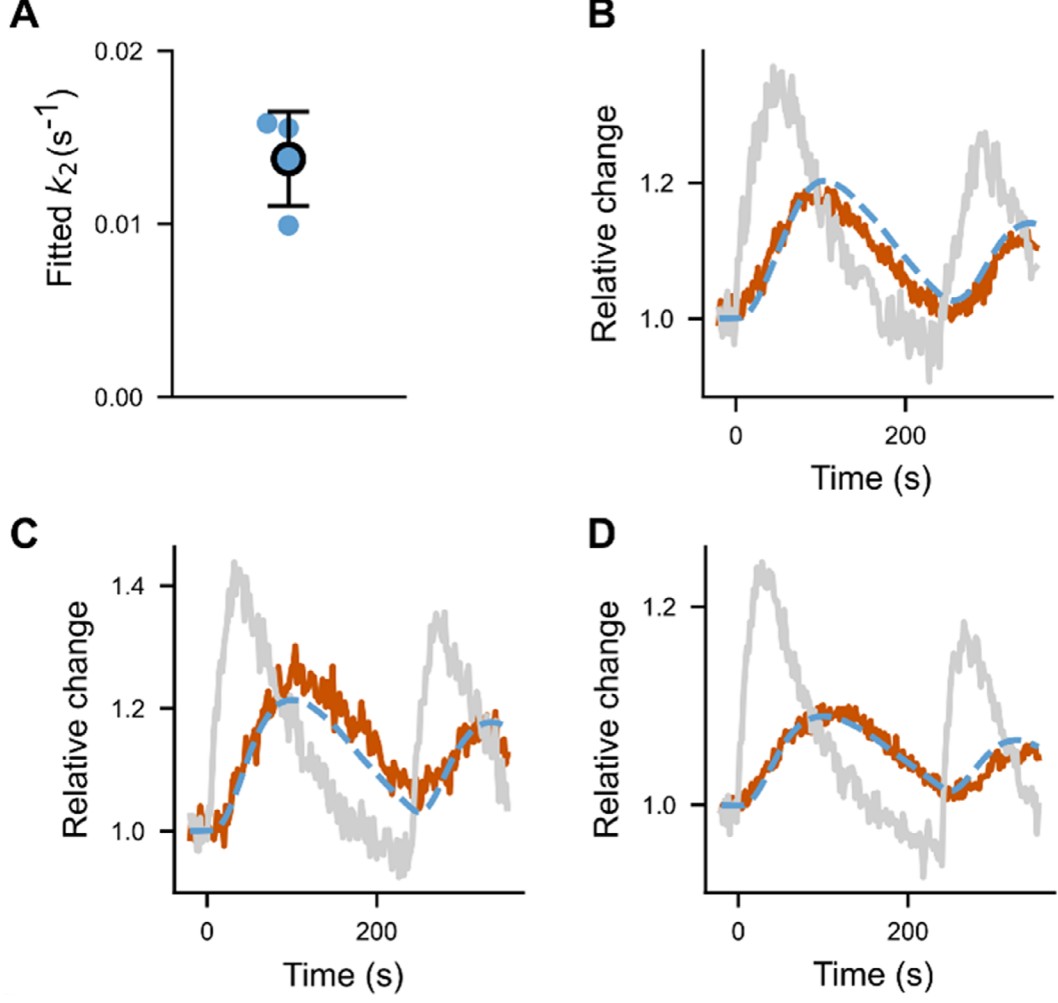

**Appendix 1—figure 2.** Fit of $k_2$. (**A**) $k_{\text{off}}$ fitted for different cells ($N$ = 3 cells). (**B–D**) The three fitted curves (fit in dotted blue, data in brown), plotted together with the normalized optoPH recruitment (gray curve).

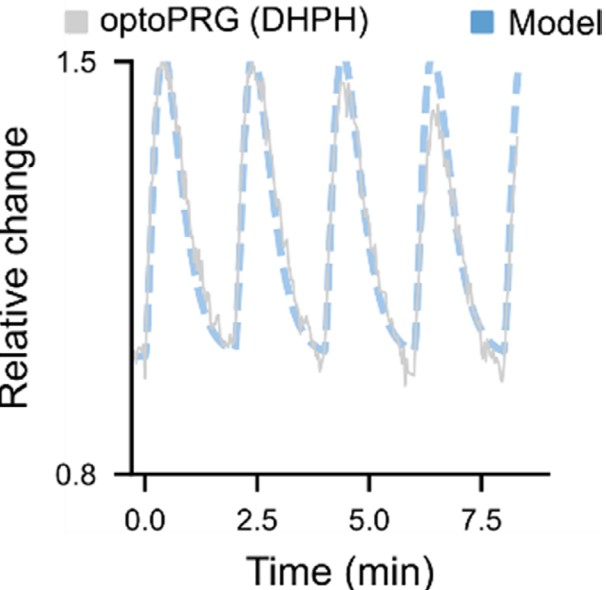

**Appendix 1—figure 3.** Fit of $f$ and $k_{off}$ for pulses of optogenetics recruitment. Pulses of light every 120 s lead to highly reproducible optogenetic recruitments that can be fitted with the model previously described. The values here are $k_{off} = 1/15s^{-1}$ and $f = 1.5$, which are the chosen values for the model.

