## [Editor Report · eLife Assessment]

This **important** study combines **compelling** experiments with optogenetic actuation and **convincing** theory to understand how signalling proteins control the switch between cell protrusion and retraction, two essential processes in single cell migration. The authors examine the importance of the basal concentration and recruitment dynamics of a guanine exchange factor (GEF) on the activity of the downstream effectors RhoA and Cdc42, which control retraction and protrusion. The experimental and theoretical evidence provides a model of RhoA's involvement in both protrusion and retraction and shows that these complex processes are highly dependent on the concentration and activity dynamics of the components.

---

## [Referee Report · Reviewer #1 (Public review)]

De Seze et al. investigated the role of guanine exchange factors (GEFs) in controlling cell protrusion and retraction. In order to causally link protein activities to the switch between the opposing cell phenotypes, they employed optogenetic versions of GEFs which can be recruited to the plasma membrane upon light exposure and activate their downstream effectors. Particularly the RhoGEF PRG could elicit both protruding and retracting phenotypes. Interestingly, the phenotype depended on the basal expression level of the optoPRG. By assessing the activity of RhoA and Cdc42, the downstream effectors of PRG, the mechanism of this switch was elucidated: at low PRG levels, RhoA is predominantly activated and leads to cell retraction, whereas at high PRG levels, both RhoA and Cdc42 are activated but PRG also sequesters the active RhoA, therefore Cdc42 dominates and triggers cell protrusion. Finally, they create a minimal model that captures the key dynamics of this protein interaction network and the switch in cell behavior.

The conclusions of this study are strongly supported by data, harnessing the power of modelling and optogenetic activation. The minimal model captures well the dynamics of RhoA and Cdc42 activation and predicts that by changing the frequency of optogenetic activation one can switch between protruding and retracting behaviour in the same cell of intermediate optoPRG level. The authors are indeed able to demonstrate this experimentally albeit with a very low number of cells. A major caveat of this study is that global changes due to PRG overexpression cannot be ruled out. Also, a quantification of absolute protein concentration, which is notoriously difficult, would be useful to put the level of overexpression here in perspective with endogenous levels. Furthermore, it remains unclear whether in cases of protein overexpression in vivo such as cancer, PRG or other GEFs can activate alternative migratory behaviours.

Previous work has implicated RhoA in both protrusion and retraction depending on the context. The mechanism uncovered here provides a convincing explanation for this conundrum. In addition to PRG, optogenetic versions of two other GEFs, LARG and GEF-H1, were used which produced either only one phenotype or less response than optoPRG, underscoring the functional diversity of RhoGEFs. The authors chose transient transfection to achieve a large range of concentration levels and, to find transfected cells at low cell density, developed a small software solution (Cell finder), which could be of interest for other researchers.

---

## [Referee Report · Reviewer #2 (Public review)]

This manuscript builds from the interesting observation that local recruitment of the DHPH domain of the RhoGEF PRG can induce local retraction, protrusion, or neither. The authors convincingly show that these differential responses are tied to the level of expression of the PRG transgene. This response depends on the Rho-binding activity of the recruited PH domain and is associated with and requires (co?)-activation of Cdc42. This begs the question of why this switch in response occurs. They use a computational model to predict that the timing of protein recruitment can dictate the output of the response in cells expressing intermediate levels and found that, "While the majority of cells showed mixed phenotypes irrespectively of the activation pattern, in few cells (3 out of 90) we were able to alternate the phenotype between retraction and protrusion several times at different places of the cell by changing the frequency while keeping the same total integrated intensity (Figure 6F and Supp Movie)."

---

## [Author Response]

The following is the authors’ response to the previous reviews.

**Reviewer #2 (Recommendations for the authors):**
While the authors have responded to most of the comments, a number of issues remain, most of which pertain to imprecise writing, as previously mentioned.

In the second revision of our manuscript, we tried our best to precise our writing.

For example, at high concentrations of PRG-GEF, the authors repeatedly state that RhoA is inhibited (including in the summary). While this may be functionally valid, it is imprecise. RhoA is activated (not inhibited), but its ability to promote contractility is impaired, presumably as a consequence of sequestration of the active GTPase by the PH domain of PRG-GEF. To put a finer point on this, the activity of RhoA•GTP is to bind to proteins that selectively bind active RhoA. One such protein the PH domain of PRG. In the case where PRG is overexpressed, RhoA•GTP binds to PRG. Due to the high concentrations of PRG in some cells, this outcompetes the ability of RhoA•GTP to bind other effectors such as formins or ROCK. However, there no strong evidence that RhoA is inhibited. The only hint of such evidence is a reduction in the biosensor for active RhoA, but this too is likely outcompeted by the overexpressed active GEF. There does not appear to be any disagreement about the mechanism, but rather a semantic difference.

We thank Reviewer #2 for emphasizing this semantic concern, which indeed requires clarification. We agree that RhoA is not chemically inactivated; rather, the protein remains active but is functionally sequestered. Our use of the term “inhibition” was intended to describe functional inhibition, consistent with the definition of inhibition as the act of reducing, preventing, or blocking a process, activity, or function. However, we recognize that this terminology could be interpreted as imprecise. To address this, we have clarified the text by explicitly referring to "functional inhibition of RhoA signaling" where appropriate, or by rewording to terms such as "competitive inhibition of RhoA effector binding" to more accurately reflect the mechanism.

Overall, the manuscript is written in a conversational style, not with the precision expected of a scientific manuscript.

We acknowledge Reviewer #2’s comment regarding the style of our manuscript. While our manuscript adopts a somewhat conversational tone, this was a deliberate choice. We believe this style helps engage the reader and facilitates understanding of our reasoning, guided by the philosophy that science is conducted by humans and should be communicated in a way that resonates with them. That said, we fully agree that this approach should not compromise scientific precision. In response to this feedback, we have revised the manuscript to ensure greater clarity and precision while maintaining the approachable style we have chosen.

To exemplify this, I provide an alternative phrasing of one such paragraph.Lines 51-62:Here, contrarily to previous optogenetic approaches, we report a serendipitous discovery where the optogenetic recruitment at the plasma membrane of GEFs of RhoA triggers both protrusion and retraction in the same cell type, polarizing the cell in opposite directions. In particular, one GEF of RhoA, PDZ-RhoGEF (PRG), also known as ARHGEF11, was most efficient in eliciting both phenotypes. We show that the outcome of the optogenetic perturbation can be predicted by the basal GEF concentration prior to activation. At high concentration, we demonstrate that Cdc42 is activated together with an inhibition of RhoA by the GEF leading to a cell protrusion. Thanks to the prediction of a minimal mathematical model, we can induce both protrusion and retraction in the same cell by modulating the frequency of light pulses. Our ability to control both phenotypes with a single protein on timescales of second provides a clear and causal demonstration of the multiplexing capacity of signaling circuits.Here, we report that the phenotypic consequences of plasma membrane recruitment of a guanine nucleotide exchange factor (GEF), PDZ-RhoGEF (PRG, aka ARHGEF11) depends on the level of expression and degree of recruitment of the GEF. At low concentrations, recruitment of PRG induces cell retraction, consistent with the expected function of a GEF for RhoA. However, at high concentrations, Cdc42 is activated, leading to cell protrusion. A minimal mathematical predicts, and experimental observations confirm, that the extent of recruitment determines the consequences of GEF recruitment. The ability of a single GEF to induce disparate outcomes demonstrates the multiplexing capacity of signaling circuits.

We thank Reviewer #2 for providing an alternative phrasing for lines 51–62. We appreciate the effort to enhance clarity and precision in this key section of the manuscript. While we agree with many aspects of the suggested revision and have incorporated several elements to improve the text, we have also retained aspects of our original phrasing that align with the overall tone and structure of the manuscript. Specifically, we have ensured that the balance between precision and accessibility is maintained while integrating the reviewer's suggestions. We hope that the revised text now addresses the concerns raised.

Key points to correct throughout the manuscript are:- overexpression of PRG does not "inhibit" RhoA.- retraction and protrusion are distinct phenotypes, they are not opposite phenotypes. One results from RhoA activation, the other results from Cdc42 activation.

Regarding the term “inhibition,” we agree with the reviewer’s point and have addressed this in our earlier comment.

Regarding the terminology of "opposite phenotypes," we believe this description is valid. While protrusion and retraction arise from distinct signaling pathways (Cdc42 activation and RhoA activation, respectively), we describe them as opposite phenotypes because they represent mutually exclusive cellular behaviors. A cell cannot protrude and retract at the same location simultaneously; instead, these behaviors represent opposing ends of the dynamic spectrum of cell morphology.

Here are some other places where editing would improve the manuscript (a noncomprehensive list).

We went through the whole manuscript to improve the scientific precision according to Reviewer #2 comment on the terminology “inhibition”.

line 15 "inhibition of RhoA by the PH domain of the GEF at high concentrations."

We modified the wording: “sequestration of active RhoA by the GEF PH domain at high concentrations”

line 51 "Here, contrarily to previous optogenetic approaches"

We removed “contrarily to previous optogenetic approaches"

line 141 "We next wonder what could differ in the activated cells that lead to the two opposite phenotypes." (the state of mind of the authors is not relevant)

As explained earlier, we made the choice to keep our writing style.

line 185 "Very surprised by this ability of one protein to trigger opposite phenotypes"

As explained earlier, we made the choice to keep our writing style.

lines 206 ff "As our optogenetic tool prevented us from using FRET biosensors because of spectral overlap, we turned to a relocation biosensor that binds RhoA in its GTP form. This highly sensitive biosensor is based on the multimeric TdTomato, whose spectrum overlaps with the RFPt fluorescent protein used for quantifying optoPRG recruitment. We thus designed a new optoPRG with iRFP, which could trigger both phenotypes *but was harder to transiently express* (?? what does this have to do with the spectral overlap), giving rise to a majority of retracting phenotype. *Looking at the RhoA biosensor*, we saw very different responses for both phenotypes (Figure 3G-I). "

We have clarified.

lines 231ff "RhoA activity shows a very different behavior: it first decays, and then rises. It seems that, adding to the well-known activation of RhoA, PRG DH-PH can also negatively regulate RhoA activity." again, RhoA activity may appear to decay, but this is a limitation of the measurements. RhoA is likely activated to the GTP-bound form. PRG is not negatively regulating RhoA activity. An activity that prevents nucleotide exchange by RhoA or accelerates its hydrolysis would constitute negative regulation of RhoA.

We modified the wording to clarify the sentence.

The attempts to quantify the degree of overexpression, though rough, should be included in the version of record. It is not clear how that estimate was generated.

The estimate of absolute concentration (switch at 200nM) was obtained by comparing fluorescent intensities of purified RFPt and cells under a spinning disk microscope while keeping the exact same acquisition settings. The whole procedure will be described in a manuscript in preparation, focused on Rac1 GEFs.